# LEARNING IN-DISTRIBUTION REPRESENTATIONS FOR ANOMALY DETECTION

## ABSTRACT

Anomaly detection involves identifying data patterns that deviate from the anticipated norm. Traditional methods struggle in high-dimensional spaces due to the curse of dimensionality. In recent years, self-supervised learning, particularly through contrastive objectives, has driven advances in anomaly detection by generating compact and discriminative feature spaces. However, vanilla contrastive learning faces challenges like class collision, especially when the In-Distribution (ID) consists primarily of normal, homogeneous data, where the lack of semantic diversity leads to increased overlap between positive and negative pairs. Existing methods attempt to address these issues by introducing hard negatives through synthetic outliers, Outlier Exposure (OE), or supervised objectives, though these approaches can introduce additional challenges. In this work, we propose the Focused In-distribution Representation Modeling (FIRM) loss, a novel multi-positive contrastive objective for anomaly detection. FIRM addresses class-collision by explicitly encouraging ID representations to be compact while promoting separation among synthetic outliers. We show that FIRM surpasses other contrastive methods in standard benchmarks, significantly enhancing anomaly detection compared to both traditional and supervised contrastive learning objectives. Our ablation studies confirm that FIRM consistently improves the quality of representations and shows robustness across a range of scoring methods. It performs particularly well in ensemble settings and benefits substantially from using OE. The code is available at `https://anonymous.4open.science/r/firm-8472/`.

## 1 INTRODUCTION

Anomaly detection is essential for identifying rare or unusual patterns in data, ensuring robustness, maintaining data quality, and preventing failures across critical applications like cybersecurity, healthcare, and autonomous systems. This task is closely related to fields such as Out-of-distribution (OOD) detection, novelty detection, and one-class classification, all of which define a specific ID to identify data points that do not conform to expected patterns under the open-world assumption (Yang et al., 2024). While both anomaly and OOD detection deal with deviations from the ID, OOD detection typically involves distinguishing between predefined classes and relies on a labeled dataset, where encouraging interclass variance in the representation space is crucial for classification. In contrast, anomaly detection focuses on identifying deviations from a singular normal class, where minimizing intraclass variance and tightly clustering ID representations is essential (Ruff et al., 2018; Ming et al., 2023), particularly when the ID is naturally homogeneous.

Recent methodologies in anomaly detection have increasingly leveraged distance-based metrics, utilizing deep representation learning extracted from models (Ruff et al., 2018; Tack et al., 2020; Sun et al., 2022; Reiss & Hoshen, 2023). These methods operate under the assumption that OOD samples are situated relatively far from the compact clusters formed by ID data. By quantifying angular separation or spatial distance between test samples and ID representations, distance-based methods distinguish between ID and OOD instances based on their relative positioning in the representation space. Their success, however, is closely tied to the quality of the learned embeddings—compact, well-separated ID representations are essential for ensuring clear boundaries between normal and anomalous data (Ming et al., 2023).

Recent progress in self-supervised learning has highlighted the efficacy of contrastive learning in learning representations across various domains, including applications such as computer vision and audio processing (Sohn, 2016; Wu et al., 2018; Chen et al., 2020a;b; He et al., 2020; Chen et al., 2020c; 2021; Tian et al., 2024). These advancements have also extended to anomaly and OOD detection. Recent studies on contrastive learning have achieved state-of-the-art results in anomaly detection and related tasks such as one-class classification and outlier detection (Sohn et al., 2021; Tack et al., 2020; Sehwag et al., 2021; Sun et al., 2022; Reiss & Hoshen, 2023). These methods rely on synthetic outliers, also referred to as virtual outliers (Du et al., 2022), which can be generated through transformations such as rotations (Golan & El-Yaniv, 2018; Hendrycks et al., 2019), adversarial training (Lee et al., 2018), or sampled in the form of OE (Hendrycks et al., 2018). Even though not necessarily from the same distribution as the actual anomalies, the synthetic outliers serve as hard negatives (Robinson et al., 2020) to the ID during contrastive training. These strategies show effectiveness in reducing the uniformity of representations (Sohn et al., 2021), a problem that occurs when learning ID representations with vanilla contrastive objectives (Sohn, 2016; Oord et al., 2018; Chen et al., 2020a) that promote a uniform distribution on the unit hypersphere (Wang & Isola, 2020a), especially in scenarios with large batch sizes.

Traditional contrastive learning training objectives maximize the similarity between semantically related instances while minimizing the similarity from less related, randomly chosen negative pairs. Applying these objectives within anomaly detection settings that rely solely on ID samples inadvertently leads to *class collision* (Arora et al., 2019). Although incorporating synthetic outliers partially addresses this issue, the foundational structure of vanilla contrastive objectives (Sohn, 2016; Oord et al., 2018), such NT-Xent (Chen et al., 2020a), which relies on a single-positive pairing strategy, continues to promote unnecessary intraclass variance among ID representations. Alternatively, Supervised Contrastive (SupCon) (Khosla et al., 2020) could be employed to reduce intraclass variance by encouraging tighter clustering of representations for both ID and synthetic outliers. However, this approach assumes homogeneity within both groups and does not account for the inherent semantic variability among synthetic outliers, as is commonly observed with OE. To overcome these limitations, we propose the FIRM loss function, a multi-positive contrastive training objective tailored for anomaly detection. The FIRM objective is designed to (1) reduce intraclass variance among ID samples, encouraging stronger alignment of ID representations in the feature space, (2) promote representation diversity among synthetic outliers, preventing mode collapse and ensuring clear distinction from ID samples. Our main contributions include:

- We introduce FIRM, a novel contrastive objective for anomaly detection, demonstrating superior performance compared to recent similar methods (Sohn et al., 2021; Tack et al., 2020) that rely on traditional contrastive learning approaches.
- We extend the applicability of FIRM to unlabeled multiclass OOD detection, demonstrating capabilities while handling non-homogeneous and multimodal ID, even on large-scale datasets like ImageNet.
- Through extensive ablation studies, we provide insights into the behavior of FIRM and its advantages over other contrastive objectives for anomaly detection. Our experiments highlight significant improvements in representation quality, particularly with OE.

## 2 LEARNING ROBUST IN-DISTRIBUTION REPRESENTATIONS

Anomaly detection aims to effectively distinguish normal (ID) samples from OOD anomalies. Given a dataset $\mathcal{D}_{\text{in}} = \{x_1, x_2, \cdots, x_N\}$ of $N$ ID samples drawn from $P_{\text{in}}$ over the input space $\mathcal{X}$, the objective is to learn an encoder $f_\theta : \mathcal{X} \to \mathbb{R}^d$ that ideally maps all ID samples to a single embedding $\mathbf{v} \in \mathbb{R}^d$, such that $f_\theta(x) = \mathbf{v}$ for all $x$, while anomalies $x'$ that diverge from $P_{\text{in}}$ are projected to distinct points such that $f_\theta(x') \notin \{\mathbf{v}\}$. While this idealized behavior may not be practically attainable, it serves as a guiding principle for designing objectives, particularly for those cases where distance-based metrics such as cosine similarity are employed wherein ID samples are concentrated in high-density areas, and OOD samples are mapped to dispersed regions, maximizing angular separation. Conventional contrastive learning objectives, such as NT-Xent, encourage excessive separation among ID samples, countering the goal of compact ID clustering. Similarly, binary approaches like SupCon, which treat all anomalies as a single class, fail to capture their inherent diversity, yielding

weaker contrastive signals during training. Moreover, having a discriminative representation space for anomalies can be beneficial for tasks requiring precise anomaly characterization, such as identifying distinct cardiac arrhythmias (Goldberger et al., 2000). Building on these observations, we introduce a contrastive training objective that incorporates a multi-positive strategy for ID samples to promote compact clustering, while leveraging synthetic outliers with single-positive pairing to enhance separation and improve the discriminative capability of anomaly detection.

**Learning through contrastive objectives.** Self-supervised learning frameworks, like contrastive representation learning, optimize a loss function to bring representations of semantically similar samples closer together while pushing dissimilar ones farther apart. For a given unlabeled sample $x_i \sim \mathcal{X}$, a stochastic data augmentation function $\alpha$ is applied to create two correlated *instances*, denoted as a positive pair $(\tilde{x}_i, \tilde{x}_{i+})$. In a minibatch of $n$ samples, the augmentation process leads to a multiview minibatch $\mathcal{B} = \{1, \ldots, 2n\}$. Within this multiview batch, each view $\tilde{x}_i$ and its counterpart $\tilde{x}_{i+}$ serve as the *anchor* and the *positive* respectively, while all other samples are treated as *negatives*. Each view within $\mathcal{B}$ is encoded via a neural network encoder $f_\theta$, parametrized by $\theta$, into representation vectors $f_\theta(\tilde{x}_i) \in \mathbb{R}^d$. The encoder's output is further transformed by a projection network $g_\psi$, parametrized by $\psi$, into a lower-dimensional space, resulting in vectors $z_i = g_\psi(f_\theta(\tilde{x}_i)) \in \mathbb{R}^{d_{\text{head}}}$ where $d_{\text{head}} < d$ (Gidaris et al., 2018). The core of the learning process is driven by a contrastive objective function (Sohn, 2016; Oord et al., 2018; Chen et al., 2020a), which encourages the maximization of the similarity between the representations of the anchor and the positive while minimizing similarity to negatives. Following this notation, the contrastive loss (Chen et al., 2020a) takes the following form:

$$\mathcal{L}_{\text{NT-Xent}}(\mathcal{B}) = -\sum_{i \in \mathcal{B}} \log \frac{\exp(z_i \cdot z_{i+}/\tau)}{\sum_{a \in \mathcal{B} \setminus \{i\}} \exp(z_i \cdot z_a/\tau)}, \tag{1}$$

where $z_i = g_\psi(f_\theta(\tilde{x}_i))/\|g_\psi(f_\theta(\tilde{x}_i))\|$ are the normalized outputs of the projection network $g_\psi$, symbol $\cdot$ denotes the dot product, and $\tau$ is a positive scalar known as the temperature parameter. Adopting the terminology proposed by Chen et al. (2020a), we refer to this formulation of the contrastive loss as the normalized temperature-scaled cross-entropy loss (NT-Xent) throughout this paper.

**Learning ID representations with synthetic outliers.** Contrastive learning relies heavily on negative samples for effective training, as evidenced by the denominator in Equation (1). Learning robust ID representations is challenging when the training data consists solely of normal samples. This limitation results in minibatches $\mathcal{B}$ consisting solely of ID samples, where negatives for any given positive pair, derived by augmenting an ID sample, are also other ID samples, which exacerbates class collision (Arora et al., 2019). To address this issue and promote clustering of ID samples' representations, the training data can be expanded through the inclusion of synthetic outliers, either generated from $P_{\text{in}}$ to closely mimic ID characteristics while lying on the low-density boundary of the ID space, or sourced from external datasets in the form of OE. These synthetic outliers, whether generated or sourced, act as hard negatives that enhance the model's discriminative capability by challenging its ability to differentiate between closely similar ID and synthetic OOD samples.

Previous works (Golan & El-Yaniv, 2018; Tack et al., 2020; Sohn et al., 2021) have shown that rotation is an effective transformation for generating synthetic outliers, offering significant gains over standard adversarial training (Lee et al., 2018; Hendrycks et al., 2019; Du et al., 2022). Although less effective as an augmentation function (Chen et al., 2020a), rotation can be repurposed to create synthetic outliers that serve as challenging negatives for ID samples. In this work, we use two approaches to generate synthetic outliers: (1) those synthetically generated from $P_{\text{in}}$, and (2) those sourced from OE (Hendrycks et al., 2018). For the first approach, we generate the synthetic outlier distribution $P_{\text{sout}}$ by applying deterministic shifting transformations, denoted as $\Omega_\gamma : \mathcal{X} \to \mathcal{X}$, where $\gamma \in K$ and $K = \{90°, 180°, 270°\}$, to samples from $\mathcal{D}_{\text{in}}$ for semantic anomaly detection. As for defect anomaly detection, we use CutPaste (Li et al., 2021a), which creates synthetic anomalies by cutting out a patch from an image and pasting it at a random location, and Natural Synthetic Anomalies (NSA) (Schlüter et al., 2022), which refines CutPaste by using Poisson image editing (Pérez et al., 2003) for more naturally blended anomalies. This results in the synthetic outlier set $\mathcal{D}_{\text{sout}} = \bigcup_{\gamma \in K} \{\Omega_\gamma(x) \mid x \in \mathcal{D}_{\text{in}}\}$. Importantly, our focus is not on comparing these synthetic

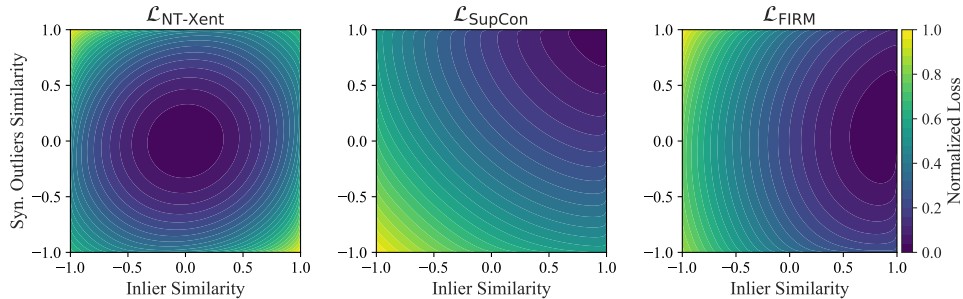

Figure 1: NT-Xent, SupCon, and FIRM loss landscapes, showing how each contrastive objective shapes the representation space and promotes either alignment or diversity among inliers and synthetic outliers.

outlier generation techniques, but rather on using these well-established methods to evaluate the effectiveness of our proposed training objective against standard ones such as NT-Xent and SupCon. For the second approach, set $\mathcal{D}_{\text{sout}}$ is sourced through OE from (Hendrycks et al., 2018). The final training data is denoted by $\mathcal{D}_{\text{in}} \cup \mathcal{D}_{\text{sout}}$. While we do not explore semi-supervised settings in this work, the proposed objective is well-suited for semi-supervised anomaly detection, akin to (Ruff et al., 2020; Sehwag et al., 2021), and can seamlessly incorporate real anomalies into $\mathcal{D}_{\text{sout}}$. Additionally, although studies show that OE tends to perform optimally when its distribution resembles closely that of $\mathcal{D}_{\text{in}}$ (Mirzaei et al., 2024), our main focus lies in evaluating our FIRM loss against established benchmarks such as NT-Xent and SupCon under identical conditions to provide a nuanced assessment of each objective's relative performance.

## 2.1 Multi-Positive Objective for In-Distribution Representation Learning

Integrating synthetic outliers into the contrastive learning framework provides essential negative samples that enhance the learning of ID representations for anomaly detection. However, employing standard objectives such as the NT-Xent loss within this setting, which inherently rely on a single-positive pairing strategy (see the numerator in Equation (1)), inadvertently still encourages intraclass variance among ID samples. This approach can hinder the model's ability to learn effective ID representations for anomaly detection, particularly when the ID is naturally homogeneous or unimodal, as it works against the goal of guiding the model toward optimal regions of the parameter space, i.e., regions where ID samples are consistently mapped to well-defined areas in the representation space, with minimal variance, and clear separation from OOD samples.

To address this challenge, we propose FIRM training objective that relies on a multi-positive contrastive learning strategy that extends the traditional contrastive loss formulation by incorporating multiple positive pairings *exclusively* for ID samples. Specifically, for each ID anchor, we identify multiple ID positives and modify the objective to align the anchor with all of these positives. In contrast, synthetic outliers are not assigned multiple positives, as they may be semantically diverse, and this diversity can be leveraged to learn a discriminative representation space.

**Training objective.** Let the ID dataset be denoted by $\mathcal{D}_{\text{in}}$, and the synthetic outliers by $\mathcal{D}_{\text{sout}}$. We assign labels such that each ID sample $x_i \in \mathcal{D}_{\text{in}}$ is assigned $y_i = 1$, and each synthetic outlier $x_j \in \mathcal{D}_{\text{sout}}$ is assigned $y_j = y_{\text{sout}}$, where $y_{\text{sout}} \neq 1$ indicates a label for synthetic outliers. Consider a multiview minibatch $\mathcal{B} = \{1, \ldots, 2n\}$, which comprises indices representing pairs of augmented instances derived from samples in $\mathcal{D}_{\text{in}} \cup \mathcal{D}_{\text{sout}}$. For each anchor $i \in \mathcal{B}$, let $\mathcal{P}(i) \subseteq \mathcal{B}$ denote the set of corresponding positive samples for $i$. In the traditional NT-Xent loss, $\mathcal{P}(i)$ is a singleton, i.e., $\mathcal{P}(i) = \{i^+\}$. To incorporate multiple positives for ID samples, we redefine the set of positive samples as $\mathcal{P}(i) = \{i^+\} \cup \{j \in \mathcal{B} \setminus \{i\} \mid y_j = 1\}$. For ID anchors $i$, $\mathcal{P}(i)$ now includes not only its paired augmented view but also all other ID samples within minibatch. For anchors $j$ corresponding to synthetic outliers, $\mathcal{P}(j)$ consists solely of the paired index $j^+$. Since synthetic outliers may exhibit significant semantic variation relative to the ID and among themselves, we retain a single-positive strategy for outliers. This approach maintains the optimization focus on minimizing intraclass variance for ID representations while preserving the model's ability to capture

and leverage the semantic diversity of the synthetic outliers during training. Following Khosla et al. (2020); Tian et al. (2024), our contrastive objective can be expressed as:

$$\mathcal{L}_{\text{FIRM}}(\mathcal{B}) = -\sum_{i \in \mathcal{B}} \frac{1}{|\mathcal{P}(i)|} \sum_{p \in \mathcal{P}(i)} \log \frac{\exp(z_i \cdot z_p/\tau)}{\sum_{a \in \mathcal{B} \setminus \{i\}} \exp(z_i \cdot z_a/\tau)}, \tag{2}$$

where $z_i$ and $\tau$ are respectively the normalized outputs of the projection network $g_\psi$ and temperature parameter. To avoid ambiguity, we reiterate on the definition of $\mathcal{P}(i)$, the set of positive samples for anchor $i \in \mathcal{B}$, as:

$$\mathcal{P}(i) = \begin{cases} \{p \in \mathcal{B} \mid y_p = y_i\} \setminus \{i\}, & \text{if } y_i = 1, \\ \{i^+\}, & \text{otherwise.} \end{cases}$$

Our approach integrates elements from both NT-Xent and SupCon losses (Khosla et al., 2020). In SupCon, the set of positives for each anchor includes all samples sharing the same label. We apply this strategy only to ID samples, leveraging their known labels following a semi-supervised anomaly detection strategy (Chandola et al., 2009). For synthetic outliers, we retain a single-positive strategy, preserving the self-supervised nature of NT-Xent and ensuring outliers remain diverse negatives for both the ID and each other.

Figure 1 illustrates the loss landscapes of NT-Xent, SupCon, and FIRM to visualize how each loss function shapes the representation space and promotes either alignment or diversity among inliers and synthetic outliers. In these plots, "Inlier Similarity" refers to the cosine similarity between the representations of two ID samples, while "Syn. Outliers Similarity" refers to the cosine similarity between two synthetic outlier representations. NT-Xent promotes diversity among ID representations by minimizing the loss when the ID similarity is low, encouraging the model to map ID samples to distinct regions of the representation space. SupCon minima lie in regions where ID representations align, but this comes at the cost of potential collapse for synthetic outliers, as it encourages unnecessary alignment between synthetic outliers. FIRM strikes a balance, encouraging alignment among ID samples while maintaining diversity among synthetic outliers.

## 2.2 DETECTION SCORE

The representations learned can be used to train methods like One-Class Support Vector Machine (OC-SVM) or Kernel Density Estimation (KDE) to score test samples to perform anomaly detection (Sohn et al., 2021), as reported in Appendix D. However, considering our objective outlined in Equation (2) encourages ID samples to align closely within the representation space, we can effectively employ distance-based score, such as the cosine similarity to the nearest neighbor from the training dataset effectively for anomaly detection. Additionally, the objective induces an increase in the norm of ID representations (Tack et al., 2020), which can also be leveraged as a distance metric.

**Detection score.** Following Reiss & Hoshen (2023); Tack et al. (2020), we employ the mean cosine similarity to the $k$ nearest neighbors as our primary detection score, denoted as:

$$s_{\text{con}}(x, \{x_m\}, k) = \frac{1}{k} \sum_{j \in N_k(x, \{x_m\})} \tilde{f}_\theta(x) \cdot \tilde{f}_\theta(x_j), \tag{3}$$

where $N_k(x, \{x_m\})$ represents the set of $k$ nearest neighbors of the sample $x$ within the set $\{x_m\}$, and $\tilde{f}_\theta(x) = f_\theta(x)/\|f_\theta(x)\|$ is the normalized feature embedding of $x$. We denote $s_{\text{con}}^*$ as the score function that includes the norm, i.e.,

$$s_{\text{con}}^*(x, \{x_m\}, k) = s_{\text{con}}(x, \{x_m\}, k) \cdot \|f_\theta(x)\|. \tag{4}$$

Note that scoring $x$ involves extracting the representations exclusively of $x$, given that the representations of the set $\{x_m\}$ can be precomputed and stored beforehand.

**Ensemble score.** We employ ensemble scores that enhance anomaly detection by incorporating transformations and augmentations during inference. Specifically, the shifting transformation score, $s_{\text{shift}}$, averages the cosine similarity across inputs rotated by $0°, 90°, 180°,$ and $270°$ degrees. The ensemble score, $s_{\text{ens}}$, extends $s_{\text{shift}}$ by including multiple random crops for each rotation degree, scaling between 0.5 and 1. This strategy enhances detection by considering rotational and spatial variations in the input. Detailed formulations of these scores are presented in Appendix B.

Table 1: Comparison of anomaly detection methods using AUROC (%). For CIFAR-10, we report the per-class results, with the final column showing the mean across all classes. We only provide the overall mean for CIFAR-100, Fashion-MNIST, and Cats-vs-Dogs. FIRM results are shown with the specified scoring function and $k = 5$. The best results with generated synthetic outliers are highlighted in bold. Additional per-class results are available in Appendix F.

(a) CIFAR-10

| Method | Network | Plane | Car | Bird | Cat | Deer | Dog | Frog | Horse | Ship | Truck | Mean |
|---|---|---|---|---|---|---|---|---|---|---|---|---|
| OC-SVM | - | 65.6 | 40.9 | 65.3 | 50.1 | 75.2 | 51.2 | 71.8 | 51.2 | 67.9 | 48.5 | 58.8 |
| DSVDD | LeNet | 61.7 | 65.9 | 50.8 | 59.1 | 60.9 | 65.7 | 67.7 | 67.3 | 75.9 | 73.1 | 64.8 |
| GEOM | WRN-16-8 | 74.7 | 95.7 | 78.1 | 72.4 | 87.8 | 87.8 | 83.4 | 95.5 | 93.3 | 91.3 | 86.0 |
| GOAD | ResNet-18 | 77.2 | 96.7 | 83.3 | 77.7 | 87.8 | 87.8 | 90.0 | 96.1 | 93.8 | 92.0 | 88.2 |
| SSD | ResNet-50 | 82.7 | 98.5 | 84.2 | 84.5 | 84.8 | 90.9 | 91.7 | 95.2 | 92.9 | 94.4 | 90.0 |
| Rot. + Trans. | WRN-16-4 | 77.5 | 96.9 | 87.3 | 80.9 | 92.7 | 90.2 | 90.9 | 96.5 | 95.2 | 93.3 | 90.1 |
| Rot. Pred. | ResNet-18 | 88.5±0.3 | 97.5±0.3 | 88.2±0.3 | 78.3±1.0 | 90.2±0.2 | 88.2±0.6 | 94.6±0.3 | 97.0±0.1 | 95.7±0.1 | 94.9±0.2 | 91.3 |
| DROC | ResNet-18 | 90.9±0.5 | 98.9±0.1 | 88.1±0.1 | 83.1±0.8 | 89.9±1.3 | 90.3±1.0 | 93.5±0.6 | 98.2±0.1 | 96.5±0.3 | 95.2±1.3 | 92.5 |
| CSI ($s_{shift}$) | ResNet-18 | – | – | – | – | – | – | – | – | – | – | 92.2 |
| CSI ($s_{ens}$) | ResNet-18 | 89.9±0.1 | 99.1±0.1 | 93.1±0.2 | 86.4±0.2 | 93.9±0.1 | 93.2±0.2 | 95.1±0.1 | 98.7±0.0 | 97.9±0.0 | 95.5±0.1 | 94.3 |
| FIRM ($s_{con}$) | ResNet-18 | 89.2±0.5 | 98.3±0.0 | 91.6±0.0 | 84.0±0.7 | 93.7±0.0 | 92.8±0.3 | 94.8±0.3 | 98.1±0.0 | 96.6±0.1 | 95.3±0.0 | 93.4 |
| FIRM ($s_{shift}$) | ResNet-18 | 92.4±0.2 | **99.2±0.0** | 93.2±0.1 | 87.9±0.2 | 94.1±0.1 | 93.9±0.2 | 96.3±0.4 | 98.7±0.1 | 97.9±0.0 | 96.3±0.0 | 95.0 |
| FIRM ($s_{ens}$) | ResNet-18 | **93.3±0.3** | **99.2±0.0** | **93.5±0.3** | **89.0±0.1** | **94.6±0.0** | **94.4±0.2** | **96.9±0.3** | **98.8±0.0** | **98.1±0.0** | **96.4±0.0** | **95.4** |
| FIRM w/ OE ($s_{con}$) | ResNet-18 | 97.7±0.1 | 99.2±0.0 | 96.1±0.0 | 92.6±0.1 | 98.2±0.0 | 96.4±0.1 | 98.9±0.0 | 98.8±0.0 | 98.9±0.0 | 99.0±0.0 | 97.6 |

(b) CIFAR-100 (superclasses)

| Method | Network | AUROC |
|---|---|---|
| GEOM | WRN-16-8 | 78.7 |
| Rot. Pred. | ResNet-18 | 84.1±0.6 |
| DROC | ResNet-18 | 86.5±0.7 |
| CSI ($s_{ens}$) | ResNet-18 | 89.6 |
| FIRM ($s_{con}$) | ResNet-18 | 87.9±0.2 |
| FIRM ($s_{shift}$) | ResNet-18 | 90.6±0.2 |
| FIRM ($s_{ens}$) | ResNet-18 | **91.0±0.2** |
| FIRM w/ OE ($s_{con}$) | ResNet-18 | 94.7±0.1 |

(c) Fashion-MNIST

| Method | Network | AUROC |
|---|---|---|
| GEOM | WRN-16-8 | 93.5 |
| GOAD | ResNet-18 | 94.1±0.5 |
| DROC | ResNet-18 | 94.5±0.4 |
| Rot. Pred. | ResNet-18 | 95.8±0.3 |
| FIRM ($s_{con}$) | ResNet-18 | **96.8±0.1** |
| FIRM ($s_{shift}$) | ResNet-18 | 96.5±0.1 |
| FIRM ($s_{ens}$) | ResNet-18 | 96.4±0.1 |
| FIRM w/ OE ($s_{con}$) | ResNet-18 | 96.2±0.1 |

(d) Cats-vs-Dogs

| Method | Network | AUROC |
|---|---|---|
| MHRot | ResNet-18 | 86.0 |
| GEOM | WRN-16-8 | 88.8 |
| Rot. Pred. | ResNet-18 | 86.4±0.6 |
| DROC | ResNet-18 | 89.6±0.3 |
| FIRM ($s_{con}$) | ResNet-18 | **90.4±0.5** |
| FIRM ($s_{shift}$) | ResNet-18 | 90.0±0.3 |
| FIRM ($s_{ens}$) | ResNet-18 | 89.7±0.5 |
| FIRM w/ OE ($s_{con}$) | – | – |

## 3 EXPERIMENTS

In this section, we present the experimental results assessing our proposed objective for anomaly detection and related tasks, including comparisons with previous works on standard benchmark datasets. Following the methodologies described in (Golan & El-Yaniv, 2018; Hendrycks et al., 2019; Bergman & Hoshen, 2020), we assess our model across several benchmarks including CIFAR-10 (Krizhevsky et al., 2009), CIFAR-100 (Krizhevsky et al., 2009), Fashion-MNIST (Xiao et al., 2017), and Cats-vs-Dogs (Elson et al., 2007). For CIFAR-100, we consider the superclass setting, where the ID consists of multiple semantically related classes, making it less homogeneous than CIFAR-10 experiments. Additionally, even though not specifically tailored for labeled OOD detection, following (Tack et al., 2020), we explore the performance of our method on the unlabeled multiclass dataset for OOD detection task using CIFAR-10 as the ID dataset, with OOD samples sourced from SVHN (Netzer et al., 2011), resized versions of LSUN, and ImageNet (Liang et al., 2018), as well as CIFAR-100 (Krizhevsky et al., 2009). To address potential dataset biases highlighted by (Tack et al., 2020), we include experiments with the corrected versions of LSUN (Fix) and ImageNet (Fix).

In all experiments, we employ the ResNet-18 architecture (He et al., 2016). Synthetic outliers come from two sources: rotations by $\{90°, 180°, 270°\}$ and OE. For OE, we utilize a curated subset of the 80 Million Tiny Images dataset, specifically the 300,000 images provided by (Hendrycks et al., 2019). This subset has been cleaned and debiased by removing images belonging to CIFAR classes, Places, LSUN classes, and those with problematic metadata. Moving forward, we refer to the experiments trained with OE as "FIRM w/ OE". We report the mean and standard deviation of evaluation metrics over five runs. Additional details about the experimental setup can be found in Appendix A.

Table 2: Image-level AUROC (%) scores on MVTec-AD dataset. NSA uses class-specific augmentations like background removal, while we apply a uniform patch blending strategy across all classes. See Table 13 for full results.

| Method | RotNet | DROC | DOCC | CutPaste | P-SVDD | U-Student | NSA | FIRM |
|---|---|---|---|---|---|---|---|---|
| Mean | 71.0±3.5 | 86.5±1.6 | 87.9 | 90.9±0.7 | 92.1 | 92.5 | **95.9±0.7** | 95.0±0.2 |

| Class | NT-Xent | SupCon | FIRM |
|---|---|---|---|
| Metal Nut | | | |
| Screw | | | |

Figure 2: t-SNE plots for the "metal nut" and "screw" classes of the MVTec-AD dataset, visualized for NT-Xent, SupCon, and FIRM loss functions. Blue points indicate normal samples, and red points indicate anomalies. Full t-SNE visualizations for all object and texture classes are included in Appendix E.

## 3.1 MAIN RESULTS

**Semantic anomaly detection** Following (Golan & El-Yaniv, 2018), we convert multiclass datasets with $C$ classes into anomaly detection tasks. Each class $c$ serves as the normal (ID) class, with others treated as anomalies (OOD). We evaluate FIRM using various scoring functions: cosine similarity ($s_{con}$), an ensemble of cosine similarities from four rotations ($s_{shift}$), and an ensemble combining crops and rotations ($s_{ens}$). Additionally, we assess "FIRM w/ OE" ($s_{con}$). For CIFAR-10 and CIFAR-100, we compare against OC-SVM (Schölkopf et al., 1999), Deep Support Vector Data Description (DSVDD) (Ruff et al., 2018), GEOM (Golan & El-Yaniv, 2018), Rot. + Trans. (Hendrycks et al., 2019), GOAD (Bergman & Hoshen, 2020), SSD (Sehwag et al., 2021), Rot. Pred. (Sohn et al., 2021), DROC (Sohn et al., 2021), and CSI (Tack et al., 2020). The results for the first four methods are sourced from (Tack et al., 2020), while those for GOAD and Rot. Pred. are from (Sohn et al., 2021). Our results, covering CIFAR-10, CIFAR-100, Fashion-MNIST, and Cats-vs-Dogs, are shown in Table 1 panels (a), (b), (c), and (d), respectively, with Cats-vs-Dogs dataset following the setup suggested in (Sohn et al., 2021) where images are resized to $64 \times 64$ pixels and to $32 \times 32$ for the other datasets. For CIFAR-10 (Table 1 (a)) shows that FIRM significantly outperforms previous methods in mean Area Under the Receiver Operating Characteristic curve (AUROC), especially under "FIRM w/ OE". FIRM exceeds DROC without ensemble and when KDE is trained on the representations (Table 7). FIRM also surpasses CSI's $s_{shift}$ ensemble and shows a strong improvement in performance with ensemble methods. Similar trends are observed in other datasets (Tables 1 (a), (b), and (c)), with FIRM consistently outperforming both DROC and CSI. However, on Fashion-MNIST and Cats-vs-Dogs, FIRM's ensemble scores do not outperform FIRM $s_{con}$. OE results for Cats-vs-Dogs are omitted due to resolution misalignment in OE data, following Sohn et al. (2021). We provide full class-wise AUROC in Appendix F.

**Defect anomaly detection** We evaluate FIRM on the MVTec anomaly detection dataset (Bergmann et al., 2021), a real-world industrial anomaly inspection benchmark. This

Table 3: AUROC (%) performance comparison for unlabeled multiclass OOD detection. The CIFAR-10 dataset is used as the ID data without labels, while various external datasets (SVHN, LSUN, ImageNet, LSUN*, ImageNet*, and CIFAR-100) serve as OOD samples. The table presents results for FIRM and "FIRM w/ OE" against multiple baseline methods. The best results with generated synthetic outliers are highlighted in bold.

| Method | In Distribution: CIFAR-10 | | | | | | Mean |
|---|---|---|---|---|---|---|---|
| | SVHN | LSUN | ImageNet | LSUN* | ImageNet* | CIFAR-100 | |
| Deep SVDD (Ruff et al., 2018) | 14.5 | - | - | - | - | 52.1 | |
| Complexity (Serrà et al., 2020) | 95.0 | - | 71.6 | - | - | 73.6 | |
| SSD (Sehwag et al., 2021) | 99.6 | - | - | - | - | 90.6 | |
| GOAD (Bergman & Hoshen, 2020) | 96.3±0.2 | 89.3±1.5 | 91.8±1.2 | 78.8±0.3 | 83.3±0.1 | 77.2±0.3 | 86.1 |
| Rot. + Trans. (Hendrycks et al., 2019) | 97.8±0.2 | 92.8±0.9 | 94.2±0.7 | 81.6±0.4 | 86.7±0.1 | 82.3±0.0 | 89.2 |
| CSI (Tack et al., 2020) | **99.8**±0.0 | 97.5±0.3 | **97.6**±0.3 | 90.3±0.3 | 93.3±0.1 | **89.2**±0.1 | 94.6 |
| FIRM | **99.8**±0.0 | **98.1**±0.1 | 95.6±0.3 | **95.8**±0.1 | **93.6**±0.1 | 88.7±0.3 | **95.2** |
| FIRM w/ OE | 99.8±0.1 | 97.3±0.2 | 93.7±0.1 | 97.3±0.1 | 95.3±0.1 | 90.1±0.1 | 95.5 |

==dataset includes natural and manufacturing defects across 10 object and 5 texture classes, with high-resolution RGB images up to $1024 \times 1024$ pixels. For training, we resize the images to $256 \times 256$, following (Schlüter et al., 2022). We compare FIRM against RotNet, DROC (Sohn et al., 2021), P-SVDD (Yi & Yoon, 2021), DOCC (Ruff et al., 2021), CutPaste, U-Student (Li et al., 2021a), and NSA (binary) (Schlüter et al., 2022). Table 2 reports the average results over object and texture classes. FIRM achieves 95.0 AUROC for image-level anomaly detection, approaching NSA's performance, despite NSA leveraging class-specific augmentations (Schlüter et al., 2022). Table 12 provides the class-wise AUROC breakdown. Notably, FIRM surpasses NT-Xent and SupCon (binary) by 5.0% and 10.7% AUROC, respectively. Representation collapse occurs for all three objectives in the "carpet" class. This issue arises due to weak augmentations; increasing the number of patches or strengthening the jitter applied to patches mitigates this collapse. However, we maintain the same shifting transformation setup to ensure a fair comparison. Figure 2 illustrates t-SNE plots of the embeddings for the "metal nut" and "screw" classes. The t-SNE plots illustrate that NT-Xent struggles to cluster ID samples and separate anomalies, SupCon exhibits poor separation between ID and anomaly clusters, while FIRM achieves a clear balance between compact clustering of ID samples and effective anomaly separation.==

**Unlabeled multiclass OOD detection**   In this setup, we follow Tack et al. (2020) for unlabeled multiclass OOD detection, treating the entire CIFAR-10 dataset as ID without labels, and using various external datasets as OOD samples. FIRM is compared against DSVDD (Ruff et al., 2018), Rot. Pred. (Sehwag et al., 2021), Complexity (Serrà et al., 2020), GOAD (Bergman & Hoshen, 2020), Rot. + Trans. (Hendrycks et al., 2019), CSI (Tack et al., 2020), and SSD (Sehwag et al., 2021), with results for GOAD and Rot. + Trans. sourced from Tack et al. (2020). Table 3 summarizes the results. FIRM outperforms other methods across most datasets, performing especially well on LSUN, LSUN*, and ImageNet*. Moreover, "FIRM w/ OE" further improves performance, highlighting the benefit of incorporating synthetic outliers, particularly on LSUN*, ImageNet*, and CIFAR-100. While OE boosts performance overall, it causes a slight drop on non-fixed LSUN and ImageNet, consistent with the observations in (Tack et al., 2020). FIRM's hyperparameters were tuned for anomaly detection rather than the unlabeled multiclass setting; further tuning, as suggested in Section 3.2, could improve performance in this task.

## 3.2 ABLATION STUDIES

==In this section, we conduct ablation studies to evaluate the performance of different training objectives, including NT-Xent, SupCon, and FIRM, across varying detection scores and synthetic outlier sources. Specifically, we examine two sources of synthetic outliers: rotations and outliers from OE, as well as the behavior of the FIRM loss function under different temperature values $\tau$ and batch sizes. For detailed results on the synthetic outliers and the temperature and batch size analysis, please refer to Appendix D.==

Table 4: Comparison of contrastive objectives (AUROC %) using the cosine similarity score ($s_{\text{con}}$) with $k = 1$ across various datasets. We evaluate NT-Xent, SupCon in both binary and multiclass settings (SupCon*), and FIRM. Results are reported for CIFAR-10, CIFAR-100 (superclass setting), Fashion-MNIST, and Cats-vs-Dogs. We present both the mean AUROC and AULC to reflect performance and convergence behavior. The relative improvement of FIRM over the baseline (NT-Xent) is shown in the bottom row.

| Loss | CIFAR-10 | | CIFAR-100 (superclass) | | FashionMNIST | | Cats-vs-Dogs | |
|---|---|---|---|---|---|---|---|---|
| | AUROC | AULC | AUROC | AULC | AUROC | AULC | AUROC | AULC |
| NT-Xent | 92.2±0.2 | 89.2±0.3 | 86.3±0.2 | 81.6±0.2 | 95.7±0.1 | 93.7±0.1 | 88.1±0.5 | 81.4±0.4 |
| SupCon | 86.5±0.2 | 64.3±2.7 | 80.7±0.9 | 60.5±2.1 | 96.4±0.1 | 87.5±4.7 | 58.2±0.1 | 51.5±0.5 |
| SupCon* | 92.5±0.2 | 90.5±0.1 | 85.9±0.4 | 82.5±0.3 | 96.6±0.1 | 95.5±0.1 | 89.2±0.1 | 85.3±0.2 |
| FIRM | **93.4±0.2** | **91.9±0.1** | **87.8±0.2** | **85.1±0.2** | **96.8±0.1** | **95.7±0.1** | **90.4±0.4** | **87.8±0.6** |
| Relative Improvement | +1.4% | +3.0% | +1.7% | +4.3% | +1.1% | +2.1% | +2.6% | +7.9% |

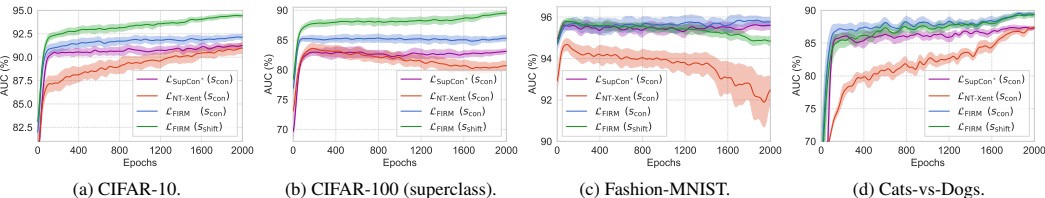

(a) CIFAR-10.  (b) CIFAR-100 (superclass).  (c) Fashion-MNIST.  (d) Cats-vs-Dogs.

Figure 3: Learning curves of contrastive objectives over 2000 training epochs on four datasets: (a) CIFAR-10, (b) CIFAR-100 (superclass), (c) Fashion-MNIST, and (d) Cats-vs-Dogs. Each plot shows the mean AUROC and variance, highlighting the convergence dynamics of NT-Xent, SupCon* (multiclass settings), and FIRM. FIRM demonstrates faster convergence and more stable performance across epochs. For a comparison between SupCon and SupCon*, refer to Appendix D.

**Contrastive objectives**  We compare FIRM's performance with that of NT-Xent and SupCon under the same training setup. SupCon is evaluated in both a binary setting (one label for ID samples and one for synthetic outliers) and a multiclass setting (SupCon*), where samples are labeled by rotation angle. The multiclass setting is not feasible for OE due to the unlabeled nature of the data. We follow the anomaly detection setup from Section 3.1, training on CIFAR-10, CIFAR-100, Fashion-MNIST, and Cats-vs-Dogs. To ensure fair comparison across loss functions, we use cosine similarity ($s_{\text{con}}$) with $k = 1$. Performance is measured by the mean AUROC over different seeds and Area Under the Learning Curve (AULC), where AULC represents the integral of the mean AUROC across training epochs. A higher AULC suggests a more effective objective, guiding the model towards better-performing regions of the parameter space. Table 4 shows that FIRM consistently outperforms NT-Xent and SupCon across all datasets. FIRM also shows significant gains in AULC, particularly on Cats-vs-Dogs, indicating higher peak performance and more efficient convergence. SupCon in the binary setting collapses in all datasets (See Appendix D for a comparison between SupCon and SupCon*). The learning curves in Figure 5 highlight the convergence dynamics, showing the mean and variance of AUROC over 2000 epochs. FIRM converges faster and more stably. On CIFAR-10 and CIFAR-100, FIRM's $s_{\text{con}}$ score plateaus around 400 epochs, while $s_{\text{shift}}$ continues to improve throughout training. Although SupCon* performs well in the multiclass setting, it fails in the binary setting, as confirmed in Table 9 (b), suggesting its effectiveness is limited to cases with well-defined (labeled) synthetic outliers, which is not guaranteed with OE.

# 4   RELATED WORK

**Self-supervised learning.**  Self-supervised learning (Gidaris et al., 2018; Hendrycks et al., 2019; Kolesnikov et al., 2019), particularly contrastive learning (Oord et al., 2018) via instance discrimination (Wu et al., 2018), has demonstrated exceptional success in visual representation learning within unsupervised settings (Chen et al., 2020a; He et al., 2020; Chen et al., 2020c; 2021). Contrastive objectives like InfoNCE (Oord et al., 2018) and NT-Xent (Chen et al., 2020a) employ a self-labeling approach, aligning positive samples in the latent space while separating negatives. Su-

pervised methods such as SupCon (Khosla et al., 2020; Tian et al., 2024) extend this by leveraging label information, enabling multiple positive samples from the same class to be clustered more tightly while enhancing class separation.

**Anomaly detection.** Traditional methods for anomaly detection span a range of approaches, from reconstruction-based techniques like Principal Component Analysis (PCA) (Jolliffe, 2002) to distance-based methods such as k-nearest neighbor (k-NN) and density estimation techniques like KDE (Parzen, 1962) and mixture models (Bishop & Nasrabadi, 2006). One-class classification algorithms, including OC-SVM (Schölkopf et al., 2001) and Support Vector Data Description (SVDD) (Tax & Duin, 2004), are also commonly employed. However, these approaches face significant challenges in high-dimensional spaces, where the curse of dimensionality (Ghosal et al., 2024) makes it difficult to model the data effectively. They rely heavily on compact representations that capture the intrinsic structure of the ID data manifold (Sohn et al., 2021).

**Representation learning for anomaly detection.** The limitations of classical approaches have driven significant interest in developing deep learning methods for anomaly detection and related tasks (Zhai et al., 2016; Chen et al., 2017; Ruff et al., 2020; Hendrycks et al., 2018; 2019; Bergman & Hoshen, 2020; Liznerski et al., 2022; Reiss & Hoshen, 2023). Most of these approaches operate in an unsupervised setting, utilizing a large corpus of unlabeled data containing normal samples (Ruff et al., 2018; Golan & El-Yaniv, 2018). However, it has been shown that incorporating additional data in the form of synthetic outliers can significantly enhance detection performance (Lee et al., 2018; Tack et al., 2020; Sohn et al., 2021; Du et al., 2022). Along similar lines, leveraging random samples from large datasets, commonly referred to as OE (Hendrycks et al., 2018), in combination with self-supervised learning (Hendrycks et al., 2019) or transfer learning (Reiss et al., 2021) has also been demonstrated to improve anomaly detection capabilities. Recent works have explored the application of contrastive learning for anomaly detection (Sohn et al., 2021; Reiss & Hoshen, 2023), particularly in OOD detection (Li et al., 2021c; Tack et al., 2020; Wang & Isola, 2020b; Sehwag et al., 2021; Li et al., 2021b; Wang & Liu, 2021; Sun et al., 2022; Ming et al., 2023) and open-set domains (Bucci et al., 2022), with most approaches employing either NT-Xent or SupCon contrastive objectives. Some works have proposed modified formulations for these tasks. Reiss & Hoshen (2023) adapts the contrastive loss for anomaly detection, improving performance but still relying on pre-trained models and offering no gains when training from scratch. Ming et al. (2023) focused on compactness and dispersion in labeled OOD detection, highlighting intraclass compactness and interclass separation. Similarly, Wang et al. (2023) proposes UniCon-HA, a contrastive learning framework combining hierarchical augmentations and re-weighting mechanisms to enhance the concentration of inliers and dispersion of outliers. Unlike UniCon-HA, which relies on complex data augmentation strategies to improve anomaly detection performance with a multi-positive contrastive loss, our work focuses on defining the characteristics of an optimal encoder for anomaly detection and aligning a contrastive learning objective with these characteristics. We aim to establish a clear connection between the design of the training objective and the optimal encoding behavior for effective anomaly detection, without relying on intricate data augmentations.

## 5 CONCLUSION

This work introduces the Focused In-distribution Representation Modeling (FIRM) loss function, designed to enhance in-distribution representation learning for robust anomaly and out-of-distribution detection. Through comprehensive experiments across multiple datasets, FIRM has proven effective in leveraging synthetic outliers to improve the discriminative capabilities of models, particularly under outlier exposure settings. Our results demonstrate the intricate interplay between various factors, including the training objective, which plays a critical role in achieving robust representations and improved model performance.

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

## A  TRAINING DETAILS

We use ResNet-18 (He et al., 2016) as the encoder network $f_\theta$ and a stack of linear, batch normalization, and ReLU, for MLP projection head $g_\phi$ consisting of 8 layers structured as $[512 \times 8, 128]$. All models are trained by minimizing the FIRM loss with a temperature of $\tau = 0.2$, except for the unlabeled OOD detection experiments, where $\tau = 0.5$ is used. We optimize the models for 2000 epochs using SGD with a momentum of 0.9. The learning rate is linearly warmed up for the first 20 epochs, then follows a cosine annealing schedule. We apply L2 weight regularization with a coefficient of 0.0003. Following Sohn et al. (2021) and our ablation studies, we set the learning rate to 0.01 and use a batch size of 32. We employ SimCLR augmentations for data augmentation, including resizing and cropping, color jitter, and Gaussian blur (Chen et al., 2020a). Note that the training recipe and hyperparameters reported herein align with the default settings provided in our publicly available code repository, which can be accessed at https://anonymous.4open.science/r/firm-8472/.

# B  ADDITIONAL SCORE FUNCTIONS

**Ensemble score.**   While employing rotation transformations to generate synthetic outliers during training, we can enhance the detection score by incorporating these transformations during inference. Specifically, the shifting transformation score averages the cosine similarity across multiple rotated input versions. Formally, the score is defined as:

$$s_{\text{shift}}(x, \{x_m\}, k) = \frac{1}{4} \sum_{\gamma \in \{0,90,180,270\}} s_{\text{con}}(R_\gamma(x), R_\gamma(\{x_m\}), k), \tag{5}$$

where $R_\gamma(x)$ denotes the rotation of sample $x$ by $\gamma$ degrees, and $R_\gamma(\{x_m\})$ denotes the set $\{x_m\}$ with each element rotated by $\gamma$ degrees. Note that this scoring function is ineffective when synthetic outliers are sourced through OE, even if rotation transformations are enforced during training (See Section 3.2).

To further improve robustness, we propose an ensemble score, $s_{\text{ens}}$, which extends the shifting transformation score by incorporating multiple random crops. For each rotation degree $\theta$, we perform ten random crops of $x$, scaling within $[0.5, 1]$, and average the scores. The ensemble score is given by:

$$s_{\text{ens}}(x, \{x_m\}, k) = \frac{1}{4} \sum_{\theta \in \{0,90,180,270\}} \frac{1}{10} \sum_{i=1}^{10} s_{\text{con}}(C_i(R_\theta(x)), R_\theta(\{x_m\}), k), \tag{6}$$

where $C_i$ denotes the $i$-th random crop applied after the rotation $R_\theta$. This ensemble strategy enhances detection by considering both rotational and spatial variations in the input.

**Center score.**   We introduce a center-based scoring function which measures the cosine similarity between the sample and the center (prototype) of the learned ID representations, defined as

$$s_{\text{center}}(x) = 1 - \cos(\phi(x), c), \tag{7}$$

where $c \in \mathbb{R}^d$ is the center of the ID representations, and $\cos$ denotes the cosine similarity between $\phi(x)$ and $c$.

# C  GENERATIVE AND DISCRIMINATIVE MODELS FOR ANOMALY DETECTION

For completeness, we provide the formulations of the classical methods KDE (Parzen, 1962) and OC-SVM (Schölkopf et al., 2001) employed in our ablation studies, which are often used for anomaly detection and related tasks. KDE is a non-parametric technique to estimate a random variable's probability density function. Following the notation from Sohn et al. (2021), the normality score using KDE with a Radial Basis Function (RBF) kernel, parameterized by $\gamma$, is formulated as follows:

$$\text{KDE}_\gamma(x) = -\frac{1}{\gamma} \log \left[ \sum_y \exp\left(-\gamma \|x - y\|^2\right) \right]. \tag{8}$$

OC-SVM solves the optimization problem by finding support vectors that describe the boundary of the ID distribution. The formulation using a linear kernel (Schölkopf et al., 2001) is given as follows:

$$\min_{w, \rho, \xi} \frac{1}{2} \|w\|^2 + \frac{1}{\nu n} \sum_{i=1}^n \xi_i - \rho \tag{9}$$

subject to

$$w^T f(x_i) \geq \rho - \xi_i, \quad \xi_i \geq 0, \quad \forall i = 1, \ldots, n, \tag{10}$$

where $f(x_i) = x_i$ in the case of the linear kernel, representing the identity feature map. The decision score is then given by:

$$s(x) = w^T x - \rho \tag{11}$$

This formulation assumes that the linear kernel $k(x, y) = x^T y$ is used, simplifying the feature mapping and the computation of the decision score.

# D  ADDITIONAL ABLATION STUDIES

**Encouraging separation for synthetic outliers.**    Here, we evaluate whether the SupCon loss alone is sufficient for learning robust representations for anomaly detection in both binary and multiclass settings on CIFAR-10. In the binary setting, SupCon encourages compactness for both classes, ID and synthetic outliers. We compare this with the multiclass setting to determine if enhancing separability among synthetic outlier representations benefits the anomaly detection task. Note that, for OE, only the binary setting is viable given the unlabeled nature of $\mathcal{D}_{\text{sout}}$ in that case. We analyze the per-class performance and convergence behavior of the SupCon loss in both binary and multiclass settings. The binary setting treats all ID samples as one class, while synthetic outliers ($\mathcal{D}_{\text{sout}}$) are considered a separate class. On the other hand, the multiclass setting, referred to as SupCon*, labels ID samples as 0, and synthetic outliers according to their rotation angles: 90, 180, and 270 degrees are labeled as 1, 2, and 3, respectively. Table 5 reports the findings for both SupCon and SupCon* using three scoring functions: $s_{\text{con}}$, the center-based $s_{\text{center}}$, and KDE. Figure 4 illustrates the learning curves for this experiment. Results indicate that SupCon in the binary setting yields suboptimal outcomes, particularly for the "Bird" and "Cat" classes, with scores of 65.8% and 63.7%, respectively, compared to 90.3% and 80.3% for SupCon* in those classes. This suggests that enforcing compact representations for synthetic outliers may lead to representation collapse. In contrast, SupCon* consistently delivers superior performance across all three scoring functions. It can be observed from Figure 4 that the SupCon approach in the binary setting demonstrates a trend toward representational collapse after some epochs.

Table 5: AUROC (%) comparison of different scoring functions on SupCon and SupCon*, where "SupCon" denotes a binary scenario where ID is a label and synthetic outliers are another, whereas "SupCon*", ID is denoted by a label, and synthetic outliers are labeled given their rotation angle.

| Loss | Score | Plane | Car | Bird | Cat | Deer | Dog | Frog | Horse | Ship | Truck | Mean |
|---|---|---|---|---|---|---|---|---|---|---|---|---|
| SupCon | $s_{\text{con}}$ | 84.8±0.0 | 97.8±0.1 | 65.8±0.5 | 63.7±0.2 | 90.5±0.2 | 88.5±0.3 | 91.9±0.1 | 97.1±0.0 | 95.1±0.1 | 94.7±0.0 | 87.0±0.1 |
| SupCon | $s_{\text{center}}$ | 83.0±0.4 | 97.1±0.1 | 61.4±0.6 | 59.4±0.1 | 89.0±0.9 | 88.2±0.1 | 90.4±0.1 | 96.4±0.0 | 94.4±0.3 | 94.1±0.1 | 85.3±0.3 |
| SupCon | KDE | 83.5±0.0 | 97.4±0.1 | 63.8±0.2 | 59.7±0.2 | 90.2±0.4 | 88.9±0.1 | 91.2±0.1 | 96.8±0.1 | 94.9±0.1 | 94.4±0.1 | 86.1±0.1 |
| SupCon* | $s_{\text{con}}$ | 86.6±0.0 | 98.2±0.1 | 90.3±0.1 | 80.3±0.4 | 92.8±0.3 | 92.2±0.3 | 94.9±0.0 | 98.1±0.1 | 95.9±0.1 | 95.5±0.0 | 92.5±0.1 |
| SupCon* | $s_{\text{center}}$ | 85.4±0.1 | 97.7±0.1 | 89.7±0.2 | 80.1±0.3 | 91.2±0.5 | 91.9±0.2 | 94.0±0.1 | 97.4±0.2 | 94.9±0.2 | 94.8±0.1 | 91.7±0.2 |
| SupCon* | KDE | 85.4±0.3 | 98.0±0.1 | 89.9±0.2 | 80.4±0.1 | 92.1±0.5 | 92.2±0.2 | 94.5±0.1 | 97.8±0.1 | 95.4±0.1 | 95.3±0.0 | 92.1±0.2 |

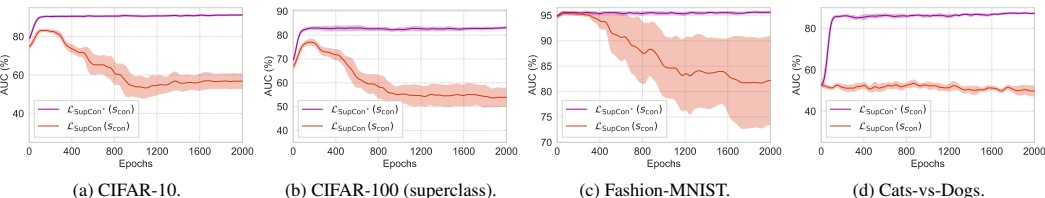

(a) CIFAR-10.          (b) CIFAR-100 (superclass).          (c) Fashion-MNIST.          (d) Cats-vs-Dogs.

Figure 4: Learning curves of SupCon (binary) and SupCon* (multiclass) contrastive objectives over 2000 training epochs on four datasets: (a) CIFAR-10, (b) CIFAR-100 (superclass), (c) Fashion-MNIST, and (d) Cats-vs-Dogs. Each plot shows the mean AUROC and variance of SupCon under two settings.

**Score functions**    We assess the impact of different scoring functions on NT-Xent, SupCon (binary and multiclass), and FIRM. The scoring functions include those from Section 2.1: $s_{\text{con}}$, $s_{\text{con}}^*$, $s_{\text{shift}}$, $s_{\text{shift}}^*$, $s_{\text{ens}}$, and $s_{\text{ens}}^*$. We also introduce a center-based scoring function, $s_{\text{center}}$, which measures cosine similarity between a sample and the center of the learned ID representations (See Appendix B for detailed formulation of $s_{\text{center}}$). To evaluate the effectiveness of ensemble scores with OE, we analyze "FIRM w/ OE (rot.)" where rotations are applied to synthetic outliers during training. Table 6 shows the results across scoring functions. FIRM improves notably with ensemble-based scores, though including norms (e.g., $s_{\text{con}}^*$, $s_{\text{ens}}^*$) does not significantly improve results. The center-based score performs well for both FIRM and SupCon in the multiclass setting (denoted as SupCon*). Results with OE suggest that applying rotations degrades the performance of $s_{\text{con}}$ and $s_{\text{center}}$ without enhancing ensemble outcomes. 'FIRM w/ OE' achieves the highest result with $s_{\text{center}}$, highlighting the importance of choosing effective scoring functions for anomaly detection.

Table 6: AUROC (%) comparison of different scoring functions on NT-Xent, SupCon binary and multiclass settings (SupCon*), and FIRM for CIFAR-10. We evaluate the performance using various scoring functions: $s_{con}$, $s_{con}^*$, $s_{shift}$, $s_{shift}^*$, $s_{ens}$, $s_{ens}^*$, and $s_{center}$ with $k = 1$.

| Loss | $s_{con}$ | $s_{con}^*$ | $s_{shift}$ | $s_{shift}^*$ | $s_{ens}$ | $s_{ens}^*$ | $s_{center}$ |
|---|---|---|---|---|---|---|---|
| NT-Xent | 92.2±0.2 | 91.9±0.5 | 92.6±0.3 | 91.6±0.3 | 92.9±0.3 | 92.2±0.3 | 84.2±1.2 |
| SupCon | 86.5±0.2 | 83.6±0.6 | 87.3±0.3 | 84.3±0.6 | 87.5±0.1 | 79.4±1.9 | 85.3±0.3 |
| SupCon* | 92.5±0.1 | 92.3±0.1 | 93.5±0.1 | 93.6±0.2 | 93.8±0.1 | 93.8±0.2 | 91.7±0.2 |
| FIRM | **93.4±0.2** | **93.4±0.2** | **95.0±0.1** | **95.0±0.1** | **95.3±0.1** | **95.4±0.1** | **92.8±0.2** |
| FIRM w/ OE (rot.) | 97.0±0.1 | 94.9±0.2 | 95.7±0.1 | 91.4±0.4 | 95.5±0.2 | 88.5±1.1 | 97.2±0.1 |
| FIRM w/ OE | 97.4±0.1 | 95.2±0.2 | 96.1±0.2 | 91.4±0.3 | 95.9±0.3 | 88.9±1.9 | 97.5±0.1 |

**Density-based anomaly score.** We also examine the performance of NT-Xent, SupCon (binary), SupCon* (multiclass), and FIRM while using KDE as a scoring function on the CIFAR-10 dataset, supplementing the distance-based methods presented in Section 2.1. The scores $KDE_{shift}$ and $KDE_{ens}$ refer to the ensemble procedures described in Equations (5) and (6), respectively. However, these scores use KDE instead of cosine similarity $s_{con}$ as the underlying scoring function. Table 7 presents the results, where FIRM also demonstrates superior performance relative to the other contrastive objectives. Comparing the results with KDE to the distance-based results displayed in Table 10, we observe a minor drop in performance using KDE, yet it demonstrates that FIRM's performance remains competitive using density-based methods. Without using ensemble methods, NT-Xent shows superior performance compared to FIRM without ensemble, particularly for classes "Plane" and "Car". When considering the ensemble scores, it is evident that FIRM benefits significantly from this approach and exhibits superior performance compared to all other objectives.

Table 7: AUROC (%) comparison of NT-Xent, SupCon, and FIRM on CIFAR-10 with KDE as the scoring function.

| Loss | Score | Plane | Car | Bird | Cat | Deer | Dog | Frog | Horse | Ship | Truck | Mean |
|---|---|---|---|---|---|---|---|---|---|---|---|---|
| NT-Xent | KDE | 92.6±0.1 | 98.9±0.0 | 90.8±0.6 | 85.3±0.1 | 92.9±0.0 | 91.5±0.1 | 95.4±0.3 | 98.1±0.6 | 96.0±0.2 | 96.3±0.0 | 93.8±0.1 |
| NT-Xent | $KDE_{shift}$ | 93.0±0.3 | 99.0±0.1 | 89.6±0.8 | 83.9±0.1 | 92.7±0.0 | 91.4±0.4 | 95.8±0.4 | 98.2±0.2 | 96.4±0.2 | 96.5±0.0 | 93.7±0.2 |
| NT-Xent | $KDE_{ens}$ | 92.9±0.1 | 99.1±0.1 | 89.9±0.8 | 84.1±0.0 | 93.0±0.1 | 91.1±0.3 | 95.9±0.5 | 98.2±0.2 | 96.3±0.2 | 96.3±0.0 | 93.7±0.2 |
| SupCon | KDE | 83.5±0.0 | 97.4±0.1 | 63.8±0.2 | 59.7±0.2 | 90.2±0.4 | 88.9±0.1 | 91.2±0.1 | 96.8±0.1 | 94.9±0.1 | 94.4±0.1 | 86.1±0.1 |
| SupCon | $KDE_{shift}$ | 84.2±0.2 | 97.2±0.0 | 65.8±1.6 | 63.5±0.7 | 90.2±0.0 | 88.8±0.1 | 92.2±0.0 | 96.3±0.1 | 94.4±0.0 | 94.7±0.0 | 86.7±0.3 |
| SupCon | $KDE_{ens}$ | 84.4±0.4 | 97.1±0.1 | 63.1±3.1 | 63.7±0.2 | 90.5±0.1 | 88.5±0.2 | 93.2±0.0 | 95.9±0.0 | 94.4±0.0 | 94.6±0.0 | 86.5±0.4 |
| SupCon* | KDE | 85.4±0.3 | 98.0±0.1 | 89.9±0.2 | 80.4±0.1 | 92.1±0.5 | 92.2±0.2 | 94.5±0.1 | 97.8±0.1 | 95.4±0.1 | 95.3±0.0 | 92.1±0.2 |
| SupCon* | $KDE_{shift}$ | 85.4±0.1 | 98.2±0.1 | 91.8±0.3 | 82.2±0.1 | 93.3±0.2 | 93.0±0.2 | 96.0±0.1 | 98.3±0.1 | 96.0±0.0 | 95.7±0.1 | 93.0±0.1 |
| SupCon* | $KDE_{ens}$ | 84.3±1.4 | 98.1±0.0 | 92.1±0.1 | 82.8±0.2 | 94.1±0.1 | 92.9±0.5 | 96.6±0.0 | 98.2±0.0 | 96.4±0.0 | 95.3±0.0 | 93.1±0.2 |
| FIRM | KDE | 88.0±0.4 | 98.2±0.0 | 91.4±0.0 | 84.1±0.6 | 93.6±0.1 | 92.8±0.3 | 94.7±0.2 | 98.0±0.0 | 96.4±0.2 | 95.1±0.1 | 93.2±0.2 |
| FIRM | $KDE_{shift}$ | 90.0±0.3 | 99.0±0.0 | 93.5±0.1 | 87.3±0.5 | 94.9±0.1 | 94.3±0.0 | 97.1±0.2 | 98.7±0.0 | 97.5±0.1 | 96.7±0.0 | 94.9±0.1 |
| FIRM | $KDE_{ens}$ | 90.9±0.3 | 99.0±0.0 | 93.9±0.2 | 88.0±0.4 | 95.5±0.0 | 94.5±0.0 | 97.5±0.2 | 98.8±0.0 | 97.5±0.1 | 96.5±0.0 | 95.2±0.1 |

**Linear Separability.** Linear classifiers are frequently used to evaluate learned representations based on the principle that effective representations should be linearly separable. Despite their general limitations, they can still deliver notable performance with high-quality representations, offering valuable insights into the quality of these embeddings (Chen et al., 2020a). In this experiment, we evaluate the quality of the learned representations for anomaly detection using CIFAR-10 while training with NT-Xent and FIRM objectives. We then used the learned representations to train a Linear One-Class Support Vector Machine (LOC-SVM). Similarly to the previously described ensemble strategy with KDE, we compute ensemble results for LOC-SVM.

Table 8 highlights the comparative performance of NT-Xent and FIRM losses using LOC-SVM on the CIFAR-10 dataset. Results indicate that FIRM, both with and without the use of ensemble methods, outperforms NT-Xent, with LOC-SVM$_{ens}$ achieving the highest mean AUROC of 95.3%. Following similar results achieved with distance-based scores reported in Table 9, NT-Xent presents better performance with and without employing ensemble methods exclusively for the "Plane", "Car", and "Truck" classes. The reasons behind FIRM's underperformance for the "Plane" and "Car" classes are analyzed in Section 3.1, which attributes to the transformation used to generate synthetic outliers involving rotations. This process does not introduce enough perceptual variation

Table 8: AUROC (%) comparison of NT-Xent and FIRM on CIFAR-10 with LOC-SVM as the scoring function.

| Loss | Score | Plane | Car | Bird | Cat | Deer | Dog | Frog | Horse | Ship | Truck | Mean |
|---|---|---|---|---|---|---|---|---|---|---|---|---|
| NT-Xent | LOC-SVM | 91.7±0.9 | 99.0±0.0 | 90.3±0.8 | 84.4±0.9 | 93.0±0.0 | 91.5±0.1 | 95.0±0.7 | 97.8±0.3 | 95.5±0.3 | 96.4±0.2 | 93.5±0.3 |
| NT-Xent | LOC-SVM$_{shift}$ | 92.6±0.2 | 99.2±0.0 | 90.5±0.1 | 83.6±0.5 | 93.0±0.1 | 92.0±0.2 | 96.0±0.7 | 98.2±0.1 | 96.8±0.1 | 97.0±0.2 | 93.9±0.2 |
| NT-Xent | LOC-SVM$_{ens}$ | 92.5±0.2 | 99.2±0.0 | 90.9±0.3 | 84.6±0.4 | 93.6±0.1 | 92.5±0.2 | 96.4±0.7 | 98.3±0.1 | 97.0±0.1 | 96.8±0.1 | 94.2±0.2 |
| FIRM | LOC-SVM | 88.4±0.1 | 98.4±0.1 | 91.6±0.1 | 85.6±0.1 | 93.3±0.0 | 93.4±0.1 | 95.2±0.3 | 98.0±0.0 | 96.6±0.0 | 95.2±0.2 | 93.6±0.1 |
| FIRM | LOC-SVM$_{shift}$ | 90.3±0.1 | 99.0±0.0 | 93.3±0.1 | 88.1±0.1 | 94.4±0.3 | 94.4±0.1 | 96.7±0.3 | 98.7±0.0 | 97.6±0.1 | 96.3±0.1 | 94.9±0.1 |
| FIRM | LOC-SVM$_{ens}$ | 91.1±0.0 | 99.0±0.0 | 93.6±0.2 | 89.2±0.1 | 95.0±0.1 | 94.9±0.1 | 97.3±0.3 | 98.7±0.0 | 97.6±0.0 | 96.4±0.2 | 95.3±0.1 |

to significantly alter the distribution for these classes, resulting in a class collision, which is exacerbated by the multi-positive strategy. This issue is not observed when using OE. Therefore, we hypothesize that with the appropriate synthetic outlier generation, the mean AUROC for LOC-SVM using FIRM could be significantly improved. Moreover, Sohn et al. (2021), using a training setup akin to ours but with the InfoNCE contrastive objective (Oord et al., 2018), report LOC-SVM results with a mean AUROC of 90.7% on CIFAR-10, while our implementation with FIRM achieves 93.6%.

**Source of Synthetic Outliers** We further investigate the impact of synthetic outliers on the performance of FIRM, NT-Xent, and SupCon. Table 9 presents results for CIFAR-10, comparing two settings: (a) synthetic outliers generated through rotations and (b) outliers from OE. In the first setting, FIRM consistently outperforms NT-Xent and SupCon, particularly excelling in complex classes like "Bird" and "Cat," though classes like "Plane" and "Car" show lower performance, possibly due to insufficient perceptual differences introduced by synthetic outliers derived from rotations. This can cause class collisions, where hard negatives overlap with ID samples, exacerbated by the multi-positive strategy. In the second setting, with synthetic outliers from OE, FIRM achieves even more significant improvements, especially in challenging classes. In the second setting, using outliers from OE, the performance issues observed in classes like "Plane" and "Car" are no longer present. These findings emphasize the importance of synthetic outlier selection, with FIRM effectively leveraging diverse and random outliers from OE to learn robust ID representations for anomaly detection.

Table 9: Per-class AUROC (%) comparison for CIFAR-10 using different sources of synthetic outliers: (a) synthetic outliers generated through rotations, and (b) synthetic outliers sourced via OE. We evaluate the performance of NT-Xent, SupCon (binary and multiclass settings), and FIRM, using the cosine similarity score ($s_{con}$) with $k = 1$. The relative improvement of FIRM over the best baseline (NT-Xent) is shown in the bottom row.

(a) Synthetic outliers generated through rotations.

| Method | Plane | Car | Bird | Cat | Deer | Dog | Frog | Horse | Ship | Truck | Mean |
|---|---|---|---|---|---|---|---|---|---|---|---|
| NT-Xent | **90.8±0.3** | **98.7±0.0** | 89.0±0.0 | 81.7±0.3 | 91.0±0.3 | 89.4±0.3 | 93.4±0.0 | 97.8±0.0 | 95.4±0.4 | 95.0±0.5 | 92.2±0.2 |
| SupCon | 85.0±0.1 | 97.7±0.0 | 65.6±0.2 | 62.0±0.6 | 89.7±0.0 | 87.8±0.0 | 91.3±0.2 | 96.8±0.1 | 95.0±0.1 | 94.3±0.2 | 86.5±0.2 |
| SupCon* | 87.0±0.0 | 98.1±0.0 | 90.4±0.2 | 80.3±0.3 | 92.8±0.3 | 92.1±0.3 | **94.9±0.0** | **98.1±0.2** | 96.0±0.1 | **95.5±0.1** | 92.5±0.2 |
| FIRM | 89.7±0.5 | 98.4±0.0 | **91.4±0.2** | **83.9±0.7** | **93.6±0.0** | **92.7±0.0** | 94.5±0.2 | **98.1±0.0** | **96.6±0.1** | 95.4±0.0 | **93.4±0.2** |
| Relative Improvement | -1.2% | -0.3% | +2.7% | +2.7% | +2.9% | +3.7% | +1.2% | +0.3% | +1.3% | +0.4% | +1.4% |

(b) Synthetic outliers via OE. "Rot." results are from (Hendrycks et al., 2019).

| Method | Plane | Car | Bird | Cat | Deer | Dog | Frog | Horse | Ship | Truck | Mean |
|---|---|---|---|---|---|---|---|---|---|---|---|
| Rot. w/ OE | 90.4 | 99.3 | 93.7 | 88.1 | 97.4 | 94.3 | 97.1 | 98.8 | 98.7 | 98.5 | 95.6 |
| NT-Xent w/ OE | 91.6±0.1 | 98.5±0.0 | 87.5±0.9 | 78.9±0.2 | 90.3±0.9 | 88.3±0.1 | 95.8±0.1 | 95.0±0.1 | 96.0±0.2 | 95.0±0.2 | 91.7±0.3 |
| SupCon w/ OE | 96.7±0.0 | 98.4±0.1 | 91.0±0.9 | 86.7±0.9 | 97.5±0.6 | 93.0±0.2 | 97.8±0.1 | 97.7±0.1 | 98.0±0.0 | 98.1±0.1 | 95.5±0.3 |
| FIRM w/ OE | **97.6±0.2** | **99.2±0.0** | **95.8±0.0** | **92.1±0.3** | **98.1±0.0** | **96.2±0.0** | **98.8±0.0** | **98.6±0.1** | **98.8±0.0** | **98.8±0.0** | **97.4±0.1** |
| Relative Improvement | +6.6% | +0.7% | +9.5% | +16.7% | +8.6% | +8.9% | +3.1% | +3.8% | +2.9% | +4.0% | +6.2% |

**Temperature and Batch Size Analysis** We assess the performance of the FIRM loss under different temperature values $\tau$ and batch sizes, focusing on four challenging CIFAR-10 classes: plane (0),

bird (2), cat (3), and deer (5). Temperatures range from $\tau \in \{0.1, 0.15, 0.2, 0.25, 0.3, 0.35, 0.4, 0.5\}$, and batch sizes from $\{16, 32, 64, 128, 256, 512, 1024, 2048\}$. To understand the impact of these hyperparameters, we evaluate scoring functions with and without norm, addressing differences observed in previous work. Figure 5 (a) and (b) show the mean and variance in AUROC across temperatures. Performance improves sharply from $\tau = 0.1$ and peaks at $\tau = 0.2$, consistent with Sohn et al. (2021). However, when norms are used in the score functions, performance peaks at $\tau = 0.4$, aligning with Tack et al. (2020). This suggests that using $\tau = 0.2$, as in our experiments, may explain lower performance when norms are included, indicating a higher temperature is needed for optimal results with norms. Figures 5 (c) and (d) show performance across batch sizes, with better results at smaller sizes, consistent Sohn et al. (2021). Performance drops with larger batch sizes, particularly when norms are included.

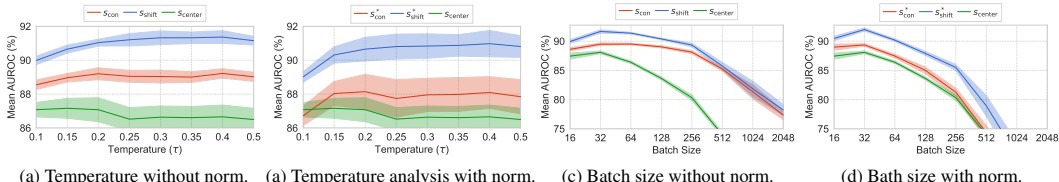

(a) Temperature without norm. (a) Temperature analysis with norm. (c) Batch size without norm. (d) Bath size with norm.

Figure 5: Analysis on the impact of varying values for $\tau$ on the performance of FIRM loss function in one-class classification.

**Results for different scoring functions and synthetic outlier sources.** We present the results for NT-Xent, SupCon, and FIRM using multiple scoring functions across CIFAR-10 classes with both generated synthetic outliers and OE. The scoring functions evaluated include those proposed in Section 2, along with the $s_{center}$ function described in Section 3.2. Scores marked with $*$ incorporate the norm, following Equation (4). Table 10 provides the results for synthetic outliers generated as described in Section 2, while Table 11 presents the results for OE from Hendrycks et al. (2018).

In the synthetic outlier experiments shown in Table 10, incorporating norms into the scoring functions within NT-Xent and SupCon configurations leads to consistently lower performance in $s_{con}$, $s_{shift}$, and $s_{ens}$ methods. This pattern is not observed with SupCon* and FIRM, where the effect of norms on performance varies. This suggests that while norms may constrain the feature space in binary scenarios like NT-Xent and SupCon, reducing their effectiveness, they can promote generalization and robustness in more complex or multiclass settings such as SupCon* and FIRM. Within NT-Xent and SupCon, $s_{ens}$ emerges as the most effective scoring function, while $s_{ens}^*$ achieves the highest AUROC scores for SupCon* and FIRM. Alternatively, the $s_{center}$ scoring function, described in Section 3.2, consistently shows the poorest performance across NT-Xent, SupCon*, and FIRM.

In the OE experiments detailed in the Table 11, including norms in the scoring functions $s_{con}$, $s_{shift}$, and $s_{ens}$ consistently degrades performance for all three loss functions. Contrarily, the $s_{center}$ scoring function, which previously showed the lowest performance in synthetic outlier scenarios for NT-Xent and FIRM, exhibits improved performance in OE, achieving the best performance for SupCon and the second best for FIRM. Notably, the most effective scoring functions in OE scenarios are $s_{con}$ for both NT-Xent and FIRM, and $s_{center}$ for SupCon, indicating a distinct interaction between the outlier data types and the scoring methods.

# E MVTEC-AD T-SNE PLOTS

This section presents the t-SNE visualizations of feature embeddings for all object and texture classes in the MVTec-AD dataset. These plots compare the embeddings generated using NT-Xent, SupCon, and FIRM. Each plot shows the distribution of normal samples (blue points) and anomalous samples (red points) in the embedding space, providing insights into the clustering and separation achieved by each loss function. Figure 6 illustrates the t-SNE plots for object classes, including "bottle," "metal nut," and "screw," among others. Similarly, Figure 7 displays the t-SNE visualizations for texture classes, such as "carpet," "grid," and "wood."

Table 10: AUROC (%) comparison for NT-Xent, SupCon, SupCon*, and FIRM losses on CIFAR-10 with synthetic outliers generated through rotations. The loss "SupCon" denotes a binary scenario where ID samples are given one label and synthetic outliers are assigned another, whereas for "SupCon*", ID samples are denoted by a label and synthetic outliers are labeled given their rotation angle. Values are reported with $k = 5$.

| Loss | Score | Plane | Car | Bird | Cat | Deer | Dog | Frog | Horse | Ship | Truck | Mean |
|---|---|---|---|---|---|---|---|---|---|---|---|---|
| NT-Xent | $s_{con}$ | 91.7±0.4 | 98.9±0.0 | 89.8±0.2 | 83.4±0.1 | 91.7±0.2 | 90.5±0.2 | 94.6±0.4 | 98.0±0.0 | 95.7±0.3 | 95.6±0.2 | 93.0±0.2 |
| NT-Xent | $s_{con}^*$ | 90.3±0.8 | 99.0±0.1 | 88.7±0.4 | 82.4±0.9 | 90.5±0.8 | 90.1±0.5 | 92.0±0.5 | 97.5±0.1 | 95.4±0.6 | 94.9±0.3 | 92.1±0.5 |
| NT-Xent | $s_{shift}$ | 92.5±0.3 | 99.1±0.0 | 88.6±0.9 | 82.8±0.2 | 91.8±0.2 | 90.3±0.2 | 95.1±0.8 | 98.1±0.2 | 96.4±0.1 | 95.9±0.1 | 93.1±0.3 |
| NT-Xent | $s_{shift}^*$ | 91.2±0.1 | 99.0±0.0 | 88.5±0.3 | 81.4±0.1 | 89.4±0.3 | 88.9±0.2 | 92.1±0.5 | 97.3±0.3 | 95.9±0.2 | 94.8±0.5 | 91.8±0.2 |
| NT-Xent | $s_{ens}$ | 92.9±0.2 | 99.1±0.0 | 89.2±0.8 | 83.2±0.2 | 92.3±0.2 | 90.0±0.1 | 95.2±0.9 | 98.1±0.2 | 96.3±0.1 | 95.8±0.1 | 93.2±0.3 |
| NT-Xent | $s_{ens}^*$ | 91.5±0.1 | 99.1±0.0 | 89.3±0.4 | 82.5±0.3 | 90.2±0.2 | 89.5±0.0 | 92.9±1.0 | 97.7±0.3 | 96.2±0.1 | 95.1±0.5 | 92.4±0.3 |
| NT-Xent | $s_{center}$ | 72.4±1.1 | 93.5±0.2 | 68.8±1.1 | 72.6±0.9 | 88.6±0.8 | 83.2±1.0 | 90.3±1.8 | 94.5±1.2 | 85.2±3.5 | 92.6±0.1 | 84.2±1.2 |
| SupCon | $s_{con}$ | 84.8±0.0 | 97.8±0.1 | 65.8±0.5 | 63.7±0.2 | 90.5±0.4 | 88.5±0.3 | 91.9±0.1 | 97.1±0.0 | 95.1±0.1 | 94.7±0.0 | 87.0±0.1 |
| SupCon | $s_{con}^*$ | 84.5±0.4 | 98.2±0.0 | 59.8±0.6 | 60.8±0.4 | 82.7±1.6 | 82.9±0.4 | 84.4±2.1 | 95.9±0.3 | 94.5±0.0 | 94.0±0.0 | 83.8±0.6 |
| SupCon | $s_{shift}$ | 86.2±0.1 | 97.6±0.0 | 68.7±0.7 | 62.7±1.1 | 91.3±0.2 | 88.7±0.1 | 92.7±0.2 | 96.9±0.3 | 95.4±0.0 | 95.0±0.0 | 87.5±0.3 |
| SupCon | $s_{shift}^*$ | 86.6±0.5 | 98.3±0.1 | 59.3±0.3 | 57.9±0.3 | 84.3±1.4 | 84.7±0.8 | 89.8±1.1 | 95.2±0.5 | 93.9±0.5 | 94.4±0.1 | 84.4±0.6 |
| SupCon | $s_{ens}$ | 86.3±0.2 | 97.5±0.0 | 68.3±0.2 | 63.7±0.1 | 91.1±0.1 | 88.1±1.0 | 93.6±0.0 | 96.3±0.0 | 95.3±0.0 | 94.8±0.0 | 87.5±0.1 |
| SupCon | $s_{ens}^*$ | 77.9±5.5 | 98.4±0.0 | 46.0±5.5 | 49.3±4.8 | 81.3±0.9 | 83.3±2.3 | 78.1±0.0 | 96.0±0.0 | 91.2±0.0 | 94.3±0.0 | 79.6±1.9 |
| SupCon | $s_{center}$ | 83.0±0.4 | 97.1±0.1 | 61.4±0.6 | 59.4±0.1 | 89.0±0.9 | 88.2±0.1 | 90.4±0.1 | 96.4±0.0 | 94.4±0.3 | 94.1±0.1 | 85.3±0.3 |
| SupCon* | $s_{con}$ | 86.6±0.0 | 98.2±0.1 | 90.3±0.1 | 80.3±0.4 | 92.8±0.3 | 92.2±0.3 | 94.9±0.0 | 98.1±0.1 | 95.9±0.1 | 95.5±0.0 | 92.5±0.1 |
| SupCon* | $s_{con}^*$ | 86.6±0.3 | 98.4±0.1 | 89.7±0.3 | 81.8±0.1 | 91.7±0.1 | 91.6±0.2 | 94.3±0.0 | 98.0±0.2 | 96.2±0.0 | 94.9±0.0 | 92.3±0.1 |
| SupCon* | $s_{shift}$ | 87.0±0.2 | 98.4±0.0 | 92.2±0.1 | 82.0±0.0 | 93.8±0.1 | 93.0±0.2 | 96.4±0.0 | 98.4±0.0 | 96.6±0.1 | 95.9±0.1 | 93.4±0.1 |
| SupCon* | $s_{shift}^*$ | 88.6±0.1 | 98.7±0.1 | 91.8±0.3 | 84.3±0.1 | 92.8±0.3 | 92.7±0.4 | 96.2±0.1 | 98.5±0.0 | 97.1±0.0 | 95.4±0.1 | 93.6±0.1 |
| SupCon* | $s_{ens}$ | 87.1±0.6 | 98.4±0.1 | 92.7±0.2 | 82.7±0.0 | 94.3±0.0 | 93.0±0.4 | 96.9±0.0 | 98.5±0.0 | 96.9±0.0 | 95.9±0.0 | 93.6±0.1 |
| SupCon* | $s_{ens}^*$ | 88.2±0.5 | 98.6±0.1 | 92.7±0.1 | 84.4±0.2 | 93.1±0.1 | 92.9±0.5 | 96.6±0.0 | 98.5±0.0 | 97.5±0.0 | 95.2±0.0 | 93.8±0.1 |
| SupCon* | $s_{center}$ | 85.4±0.1 | 97.7±0.1 | 89.7±0.2 | 80.1±0.3 | 91.2±0.5 | 91.9±0.2 | 94.0±0.1 | 97.4±0.2 | 94.9±0.2 | 94.8±0.1 | 91.7±0.2 |
| FIRM | $s_{con}$ | 89.2±0.5 | 98.3±0.0 | 91.6±0.0 | 84.0±0.7 | 93.7±0.0 | 92.8±0.3 | 94.8±0.3 | 98.1±0.0 | 96.6±0.1 | 95.3±0.0 | 93.4±0.2 |
| FIRM | $s_{con}^*$ | 89.0±0.2 | 98.5±0.1 | 91.4±0.2 | 85.4±0.1 | 92.9±0.1 | 93.2±0.2 | 94.7±0.4 | 97.9±0.0 | 96.6±0.0 | 95.0±0.2 | 93.5±0.2 |
| FIRM | $s_{shift}$ | 91.9±0.0 | 99.1±0.0 | 93.3±0.1 | 87.0±0.2 | 95.0±0.0 | 94.0±0.0 | 97.0±0.1 | 98.8±0.0 | 97.7±0.0 | 96.7±0.0 | 95.1±0.0 |
| FIRM | $s_{shift}^*$ | 92.4±0.2 | 99.2±0.0 | 93.2±0.0 | 87.9±0.1 | 94.1±0.1 | 94.0±0.0 | 96.3±0.4 | 98.7±0.1 | 97.9±0.0 | 96.3±0.0 | 95.0±0.1 |
| FIRM | $s_{ens}$ | 92.6±0.0 | 99.2±0.0 | 93.9±0.1 | 87.6±0.1 | 95.4±0.0 | 94.2±0.0 | 97.4±0.1 | 98.8±0.0 | 97.7±0.0 | 96.5±0.0 | 95.3±0.0 |
| FIRM | $s_{ens}^*$ | 93.3±0.3 | 99.2±0.0 | 93.5±0.3 | 89.0±0.1 | 94.6±0.0 | 94.4±0.2 | 96.9±0.3 | 98.8±0.0 | 98.1±0.0 | 96.4±0.0 | 95.4±0.1 |
| FIRM | $s_{center}$ | 86.5±0.0 | 98.1±0.0 | 90.2±0.0 | 83.5±0.6 | 93.1±0.2 | 92.7±0.4 | 95.0±0.2 | 97.9±0.0 | 96.0±0.2 | 94.8±0.1 | 92.8±0.2 |

Table 11: AUROC (%) comparison for NT-Xent, SupCon, and FIRM losses on CIFAR-10 with Outlier Exposure (OE), where "SupCon" denotes a binary scenario where ID samples are given one label and synthetic outliers are assigned another.

| Loss | Score | Plane | Car | Bird | Cat | Deer | Dog | Frog | Horse | Ship | Truck | Mean |
|---|---|---|---|---|---|---|---|---|---|---|---|---|
| NT-Xent w/ OE | $s_{con}$ | 92.5±0.1 | 98.7±0.0 | 88.0±0.8 | 80.5±0.4 | 91.4±0.6 | 90.2±0.4 | 96.5±0.0 | 96.2±0.0 | 96.2±0.1 | 95.7±0.2 | 92.6±0.3 |
| NT-Xent w/ OE | $s_{con}^*$ | 88.3±0.9 | 98.0±0.1 | 88.1±0.5 | 75.0±0.5 | 90.5±0.1 | 85.0±0.1 | 95.1±0.2 | 94.4±0.1 | 94.8±0.1 | 92.9±0.0 | 90.2±0.2 |
| NT-Xent w/ OE | $s_{shift}$ | 90.7±0.1 | 97.5±0.1 | 84.8±0.1 | 78.3±0.9 | 88.7±0.1 | 87.9±0.1 | 94.8±0.1 | 94.3±0.1 | 93.2±0.2 | 93.7±0.1 | 90.4±0.2 |
| NT-Xent w/ OE | $s_{shift}^*$ | 82.1±1.5 | 94.8±0.2 | 82.6±0.1 | 69.4±1.8 | 84.2±1.7 | 78.8±0.2 | 91.8±0.7 | 90.4±0.5 | 85.8±0.3 | 84.4±0.3 | 84.4±0.7 |
| NT-Xent w/ OE | $s_{ens}$ | 89.9±0.0 | 97.2±0.0 | 84.2±0.0 | 77.7±0.9 | 88.5±0.1 | 86.9±0.5 | 94.5±0.1 | 93.8±0.3 | 93.0±0.2 | 93.1±0.1 | 89.9±0.2 |
| NT-Xent w/ OE | $s_{ens}^*$ | 71.5±0.8 | 92.7±0.8 | 82.3±0.2 | 70.4±1.9 | 82.2±3.8 | 74.1±4.3 | 83.4±3.6 | 88.0±0.3 | 79.9±5.8 | 82.1±2.9 | 80.7±2.4 |
| NT-Xent w/ OE | $s_{center}$ | 88.2±1.3 | 98.2±0.1 | 75.6±0.5 | 72.5±0.7 | 88.8±2.8 | 86.1±0.2 | 96.7±0.0 | 96.6±0.6 | 96.1±0.2 | 94.7±0.0 | 89.4±0.6 |
| SupCon w/ OE | $s_{con}$ | 97.0±0.0 | 98.8±0.1 | 92.1±1.1 | 88.6±0.7 | 97.7±0.5 | 93.7±0.1 | 98.3±0.0 | 98.1±0.0 | 98.7±0.1 | 98.6±0.0 | 96.2±0.3 |
| SupCon w/ OE | $s_{con}^*$ | 93.2±0.9 | 99.0±0.1 | 84.1±0.1 | 81.9±1.2 | 93.8±1.6 | 91.8±1.1 | 96.3±0.1 | 96.2±0.5 | 95.2±0.2 | 96.1±0.1 | 92.8±0.6 |
| SupCon w/ OE | $s_{shift}$ | 96.1±0.1 | 97.9±0.5 | 89.3±0.6 | 84.2±0.9 | 96.5±1.0 | 91.4±0.3 | 97.3±0.2 | 97.1±0.3 | 98.1±0.2 | 97.9±0.0 | 94.6±0.4 |
| SupCon w/ OE | $s_{shift}^*$ | 88.8±1.4 | 98.5±0.2 | 74.9±0.7 | 78.0±2.9 | 88.4±1.4 | 88.1±1.5 | 93.8±0.5 | 93.2±0.0 | 88.4±2.2 | 93.9±0.7 | 88.6±1.1 |
| SupCon w/ OE | $s_{ens}$ | 96.5±0.0 | 97.7±0.9 | 89.6±0.8 | 83.9±0.4 | 97.0±0.8 | 90.6±1.3 | 97.7±0.1 | 97.2±0.4 | 97.7±0.6 | 97.8±0.1 | 94.6±0.6 |
| SupCon w/ OE | $s_{ens}^*$ | 83.8±0.1 | 93.1±5.3 | 68.4±1.8 | 77.8±1.3 | 69.9±7.9 | 87.1±0.3 | 88.1±2.1 | 90.4±0.2 | 58.2±12.6 | 93.3±1.5 | 81.0±3.3 |
| SupCon w/ OE | $s_{center}$ | 96.7±0.1 | 98.9±0.0 | 92.9±0.5 | 89.8±0.1 | 97.5±0.4 | 94.3±0.0 | 98.3±0.0 | 98.2±0.0 | 98.9±0.1 | 98.6±0.1 | 96.4±0.1 |
| FIRM w/ OE | $s_{con}$ | 97.7±0.1 | 99.2±0.0 | 96.1±0.0 | 92.6±0.1 | 98.2±0.0 | 96.4±0.1 | 98.9±0.0 | 98.8±0.0 | 98.9±0.0 | 99.0±0.0 | 97.6±0.0 |
| FIRM w/ OE | $s_{con}^*$ | 95.5±0.4 | 98.9±0.0 | 93.0±0.7 | 87.5±0.4 | 96.6±0.0 | 94.3±0.0 | 98.3±0.2 | 96.5±0.3 | 97.6±0.2 | 96.8±0.0 | 95.5±0.2 |
| FIRM w/ OE | $s_{shift}$ | 96.7±0.0 | 98.7±0.0 | 93.6±0.3 | 88.7±0.4 | 97.3±0.1 | 94.7±0.5 | 98.5±0.1 | 97.8±0.1 | 97.9±0.0 | 98.2±0.1 | 96.2±0.2 |
| FIRM w/ OE | $s_{shift}^*$ | 91.9±0.4 | 98.2±0.1 | 86.4±1.3 | 81.9±0.1 | 92.0±0.0 | 89.9±0.1 | 97.1±0.2 | 91.4±0.4 | 94.6±0.2 | 93.2±0.2 | 91.7±0.3 |
| FIRM w/ OE | $s_{ens}$ | 96.6±0.2 | 98.5±0.0 | 93.6±0.3 | 88.7±0.7 | 97.3±0.2 | 94.4±0.5 | 98.5±0.2 | 97.5±0.1 | 97.6±0.1 | 97.9±0.1 | 96.1±0.2 |
| FIRM w/ OE | $s_{ens}^*$ | 88.9±3.3 | 93.9±0.1 | 86.3±1.8 | 83.7±0.6 | 91.5±0.2 | 88.5±2.4 | 96.6±1.0 | 90.9±0.3 | 87.6±7.0 | 87.5±0.9 | 89.5±1.8 |
| FIRM w/ OE | $s_{center}$ | 97.4±0.1 | 99.3±0.0 | 95.9±0.1 | 92.5±0.2 | 97.9±0.0 | 96.3±0.0 | 99.0±0.1 | 99.0±0.0 | 99.0±0.1 | 98.9±0.0 | 97.5±0.1 |

# F  PER-CLASS RESULTS

In this section, we report the per-class results of the anomaly experiments for datasets CIFAR-10, CIFAR-100 (superclass setting), Fashion-MNIST, and Cats-vs-Dogs. Table 14, 15, 16, and 17

Table 12: AUROC (%) comparison for NT-Xent, SupCon, and FIRM losses on MVTec-AD with synthetic outliers generated through CutPaste. The loss "SupCon" denotes a binary scenario where ID samples are given one label and synthetic outliers are assigned another.

| Loss | NT-Xent | SupCon | FIRM |
|---|---|---|---|
| Score | $s_{con}$ | $s_{con}$ | $s_{con}$ |
| Bottle | **100.0±0.0** | **100.0±0.0** | **100.0±0.0** |
| Cable | 97.4±0.6 | 77.0±0.6 | **97.4±0.3** |
| Capsule | 80.7±1.6 | 73.6±0.5 | **92.8±0.2** |
| Hazelnut | 84.9±4.3 | 95.7±0.7 | **96.5±0.1** |
| Metal Nut | 95.9±0.4 | 80.8±2.7 | **98.6±0.2** |
| Pill | 90.0±0.1 | **97.0±0.0** | 92.8±0.6 |
| Screw | 72.7±10.7 | 39.8±6.4 | **96.7±0.5** |
| Toothbrush | 98.6±0.3 | **100.0±0.0** | **100.0±0.0** |
| Transistor | **94.3±1.0** | 81.8±9.3 | 93.1±0.3 |
| Zipper | 99.8±0.2 | **100.0±0.0** | **100.0±0.0** |
| Carpet | 65.4±6.2 | 67.6±8.0 | **75.2±0.3** |
| Grid | 94.6±0.8 | 85.5±1.0 | **100.0±0.0** |
| Leather | 88.4±0.8 | 86.0±0.4 | **93.8±0.8** |
| Tile | 99.8±0.1 | 83.6±1.8 | **100.0±0.0** |
| Wood | 87.2±4.7 | **96.8±1.1** | 87.5±0.3 |
| Mean | 90.0±2.1 | 84.3±2.2 | **95.0±0.2** |

Table 13: Image-level AUROC (%) for MVTec AD dataset. Results for RotNet and DROC are from (Sohn et al., 2021), P-SVDD from (Yi & Yoon, 2021), DOCC from (Ruff et al., 2021), CutPaste and U-Student from (Li et al., 2021a), and NSA (binary) from (Schlüter et al., 2022). FIRM results are reported for $s_{con}$.

| Method | RotNet | DROC | DOCC | CutPaste | P-SVDD | U-Student | NSA | FIRM |
|---|---|---|---|---|---|---|---|---|
| Bottle | - | - | 99.6 | 99.2±0.2 | 98.6 | 96.7 | 97.6±0.2 | **100.0±0.0** |
| Cable | - | - | 90.9 | 87.1±0.8 | 90.3 | 82.3 | 92.1±2.4 | **97.4±0.3** |
| Capsule | - | - | 91.0 | 87.9±0.7 | 76.7 | 92.8 | **93.2±0.8** | 92.8±0.2 |
| Hazelnut | - | - | 95.0 | 91.3±0.6 | 92.0 | 91.4 | 93.5±1.9 | **96.5±0.1** |
| Metal Nut | - | - | 85.2 | 96.8±0.5 | 94.0 | 94.0 | **99.4±0.3** | 98.6±0.2 |
| Pill | - | - | 80.4 | 93.4±0.9 | 86.1 | 86.7 | **97.0±0.9** | 92.8±0.6 |
| Screw | - | - | 86.9 | 93.4±0.9 | 81.3 | 87.4 | 90.3±1.2 | **96.7±0.5** |
| Toothbrush | - | - | 96.4 | 99.2±0.2 | **100.0** | 98.6 | **100.0±0.0** | **100.0±0.0** |
| Transistor | - | - | 90.8 | **96.4±0.7** | 91.5 | 83.6 | 93.5±0.9 | 93.1±0.3 |
| Zipper | - | - | 92.4 | 99.4±0.1 | 97.9 | 95.8 | 99.8±0.1 | **100.0±0.0** |
| Carpet | - | - | 90.6 | 67.9±1.8 | 92.9 | **95.3** | 85.6±7.6 | 75.2±0.3 |
| Grid | - | - | 52.4 | 99.9±0.1 | 94.6 | 98.7 | 99.9±0.1 | **100.0±0.0** |
| Leather | - | - | 78.3 | 99.7±0.1 | 90.9 | 93.4 | **99.9±0.1** | 93.8±0.8 |
| Tile | - | - | 96.5 | 95.9±1.0 | 97.8 | 95.8 | 99.7±0.2 | **100.0±0.0** |
| Wood | - | - | 91.6 | 94.9±0.5 | 96.5 | 95.5 | **96.7±1.2** | 87.5±0.3 |
| Mean | 71.0±3.5 | 86.5±1.6 | 87.9 | 90.9±0.7 | 92.1 | 92.5 | **95.9±0.7** | 95.0±0.2 |

present results for dataset CIFAR-10, CIFAR-100, Fashion-MNIST, and Cats-vs-Dogs, respectively. Results are reported with $k = 5$.

Table 14: AUROC (%) comparison across different contrastive objectives for CIFAR-10.

| Label | NT-Xent | SupCon | SupCon* | FIRM | FIRM | FIRM |
|---|---|---|---|---|---|---|
| Score | $s_{\text{con}}$ | $s_{\text{con}}$ | $s_{\text{con}}$ | $s_{\text{con}}$ | $s_{\text{shift}}$ | $s_{\text{ens}}$ |
| 0 | 91.7±0.4 | 84.8±0.0 | 86.6±0.0 | 89.2±0.5 | 91.9±0.0 | **92.6±0.0** |
| 1 | 98.9±0.0 | 97.8±0.1 | 98.2±0.1 | 98.3±0.0 | 99.1±0.0 | **99.2±0.0** |
| 2 | 89.8±0.2 | 65.8±0.5 | 90.3±0.1 | 91.6±0.0 | 93.3±0.1 | **93.9±0.1** |
| 3 | 83.4±0.1 | 63.7±0.2 | 80.3±0.4 | 84.0±0.7 | 86.9±0.1 | **87.6±0.1** |
| 4 | 91.7±0.2 | 90.5±0.2 | 92.8±0.3 | 93.7±0.0 | 95.0±0.0 | **95.4±0.0** |
| 5 | 90.5±0.2 | 88.5±0.3 | 92.2±0.3 | 92.8±0.3 | 94.0±0.0 | **94.2±0.0** |
| 6 | 94.6±0.4 | 91.9±0.1 | 94.9±0.0 | 94.8±0.3 | 97.0±0.1 | **97.4±0.1** |
| 7 | 98.0±0.0 | 97.1±0.0 | 98.1±0.1 | 98.1±0.0 | 98.8±0.0 | **98.8±0.0** |
| 8 | 95.7±0.3 | 95.1±0.1 | 95.9±0.1 | 96.6±0.1 | 97.7±0.0 | **97.7±0.0** |
| 9 | 95.6±0.2 | 94.7±0.0 | 95.5±0.0 | 95.3±0.0 | **96.7±0.0** | 96.5±0.0 |
| mean | 93.0±0.2 | 87.0±0.2 | 92.5±0.2 | 93.4±0.2 | 95.0±0.1 | **95.3±0.0** |

Table 15: AUROC (%) comparison across different contrastive objectives for CIFAR-100.

| Label | NT-Xent | SupCon | SupCon* | FIRM | FIRM | FIRM |
|---|---|---|---|---|---|---|
| Score | $s_{\text{con}}$ | $s_{\text{con}}$ | $s_{\text{con}}$ | $s_{\text{con}}$ | $s_{\text{shift}}$ | $s_{\text{ens}}$ |
| 0 | 84.8±0.1 | 78.6±0.4 | 81.3±0.2 | 83.9±0.2 | 87.1±0.0 | **87.2±0.5** |
| 1 | **86.9±0.1** | 75.6±0.0 | 78.9±0.4 | 82.9±0.0 | 86.6±0.3 | 86.7±0.3 |
| 2 | **93.4±0.3** | 79.3±1.1 | 85.1±0.9 | 89.0±0.1 | 93.0±0.3 | 93.1±0.4 |
| 3 | 88.3±0.5 | 72.6±10.6 | 88.9±0.2 | 91.1±0.2 | **92.5±0.4** | 91.5±0.7 |
| 4 | **92.8±0.3** | 81.7±0.1 | 87.5±0.5 | 89.0±0.0 | 92.6±0.2 | 92.7±0.1 |
| 5 | 83.6±0.7 | 57.2±0.8 | 77.6±0.6 | 82.8±0.1 | 86.0±0.2 | **86.5±0.2** |
| 6 | 82.2±0.3 | 89.5±0.5 | 90.8±0.0 | 92.4±0.5 | 93.9±0.2 | **94.3±0.1** |
| 7 | **87.2±0.2** | 70.8±1.1 | 75.5±0.6 | 77.8±0.7 | 84.7±0.0 | 85.4±0.4 |
| 8 | 86.8±0.1 | 87.0±0.1 | 89.1±0.5 | 89.5±0.0 | 91.5±0.2 | **92.6±0.0** |
| 9 | 93.4±0.3 | 93.0±0.3 | 93.4±0.2 | 94.0±0.0 | 95.5±0.0 | **95.9±0.0** |
| 10 | 87.3±1.0 | 87.9±0.7 | 88.5±0.3 | 88.9±0.1 | 90.7±0.0 | **90.9±0.0** |
| 11 | 86.2±0.3 | 86.6±0.2 | 88.7±0.3 | 89.6±0.2 | **91.3±0.2** | 91.3±0.0 |
| 12 | 85.3±0.1 | 86.1±0.2 | 87.4±0.0 | 89.0±0.2 | 90.9±0.1 | **91.4±0.0** |
| 13 | 81.8±0.9 | 63.8±0.4 | 70.2±1.6 | 76.7±0.6 | 81.7±0.1 | **82.9±0.0** |
| 14 | 92.4±0.2 | 92.0±0.1 | 93.2±0.2 | 94.8±0.2 | 96.2±0.3 | **96.7±0.2** |
| 15 | 76.8±0.2 | 73.4±0.1 | 76.0±0.2 | 78.2±0.2 | 81.9±0.3 | **82.6±0.2** |
| 16 | 81.6±0.3 | 66.6±1.4 | 81.9±0.6 | 83.9±0.3 | 85.9±0.3 | **86.4±0.4** |
| 17 | 97.4±0.0 | 95.3±0.2 | 96.1±0.0 | 96.4±0.0 | 98.1±0.1 | **98.4±0.0** |
| 18 | 93.6±0.0 | 93.1±0.4 | 94.2±0.1 | 94.8±0.1 | 96.1±0.2 | **96.4±0.1** |
| 19 | 93.0±0.0 | 92.3±0.2 | 92.8±0.0 | 93.7±0.1 | 96.0±0.0 | **96.3±0.1** |
| mean | 87.7±0.3 | 81.1±0.9 | 85.8±0.4 | 87.9±0.2 | 90.6±0.2 | **91.0±0.2** |

Table 16: AUROC (%) comparison across different contrastive objectives for Fashion-MNIST.

| Label | NT-Xent | SupCon | SupCon* | FIRM | FIRM | FIRM |
|---|---|---|---|---|---|---|
| Score | $s_{\text{con}}$ | $s_{\text{con}}$ | $s_{\text{con}}$ | $s_{\text{con}}$ | $s_{\text{shift}}$ | $s_{\text{ens}}$ |
| 0 | 94.6±0.1 | **96.6±0.1** | 96.4±0.0 | 96.3±0.0 | 96.0±0.1 | 96.0±0.0 |
| 1 | 99.4±0.0 | 99.7±0.0 | 99.8±0.0 | 99.8±0.0 | **99.8±0.0** | 99.8±0.0 |
| 2 | 94.8±0.0 | 95.4±0.2 | 95.5±0.1 | 95.3±0.2 | **95.5±0.0** | 95.3±0.1 |
| 3 | 94.3±0.2 | 96.5±0.3 | **96.6±0.4** | 96.5±0.2 | 95.6±0.0 | 96.0±0.0 |
| 4 | 92.7±0.1 | 95.0±0.0 | 94.9±0.0 | **95.5±0.0** | 93.7±0.1 | 93.1±0.2 |
| 5 | 95.8±0.0 | 97.0±0.2 | 98.1±0.1 | **98.6±0.1** | 97.1±0.0 | 96.8±0.0 |
| 6 | **89.0±0.1** | 86.1±0.3 | 87.5±0.4 | 87.7±0.1 | 88.7±0.2 | 88.4±0.1 |
| 7 | 98.8±0.1 | 99.5±0.0 | 99.5±0.0 | **99.6±0.0** | 99.2±0.0 | 99.3±0.0 |
| 8 | 99.0±0.1 | 99.1±0.0 | 99.5±0.0 | 99.5±0.1 | 99.7±0.0 | **99.7±0.0** |
| 9 | 99.3±0.0 | 99.4±0.1 | 99.2±0.1 | **99.5±0.0** | 99.3±0.0 | 99.3±0.0 |
| mean | 95.8±0.1 | 96.4±0.1 | 96.7±0.1 | **96.8±0.1** | 96.5±0.1 | 96.4±0.1 |

Table 17: AUROC (%) comparison across different contrastive objectives for Cats-vs-Dogs.

| Label | NT-Xent | SupCon | SupCon* | FIRM | FIRM | FIRM |
|---|---|---|---|---|---|---|
| Score | $s_{\text{con}}$ | $s_{\text{con}}$ | $s_{\text{con}}$ | $s_{\text{con}}$ | $s_{\text{shift}}$ | $s_{\text{ens}}$ |
| 0 | **91.2±0.0** | 57.3±0.7 | 87.9±0.2 | 89.8±0.7 | 89.6±0.3 | 89.8±0.3 |
| 1 | 85.0±0.3 | 59.1±0.3 | 90.4±0.1 | **91.1±0.2** | 90.3±0.3 | 90.5±0.2 |
| mean | 88.1±0.2 | 58.2±0.5 | 89.2±0.1 | **90.4±0.5** | 89.9±0.3 | 90.2±0.3 |

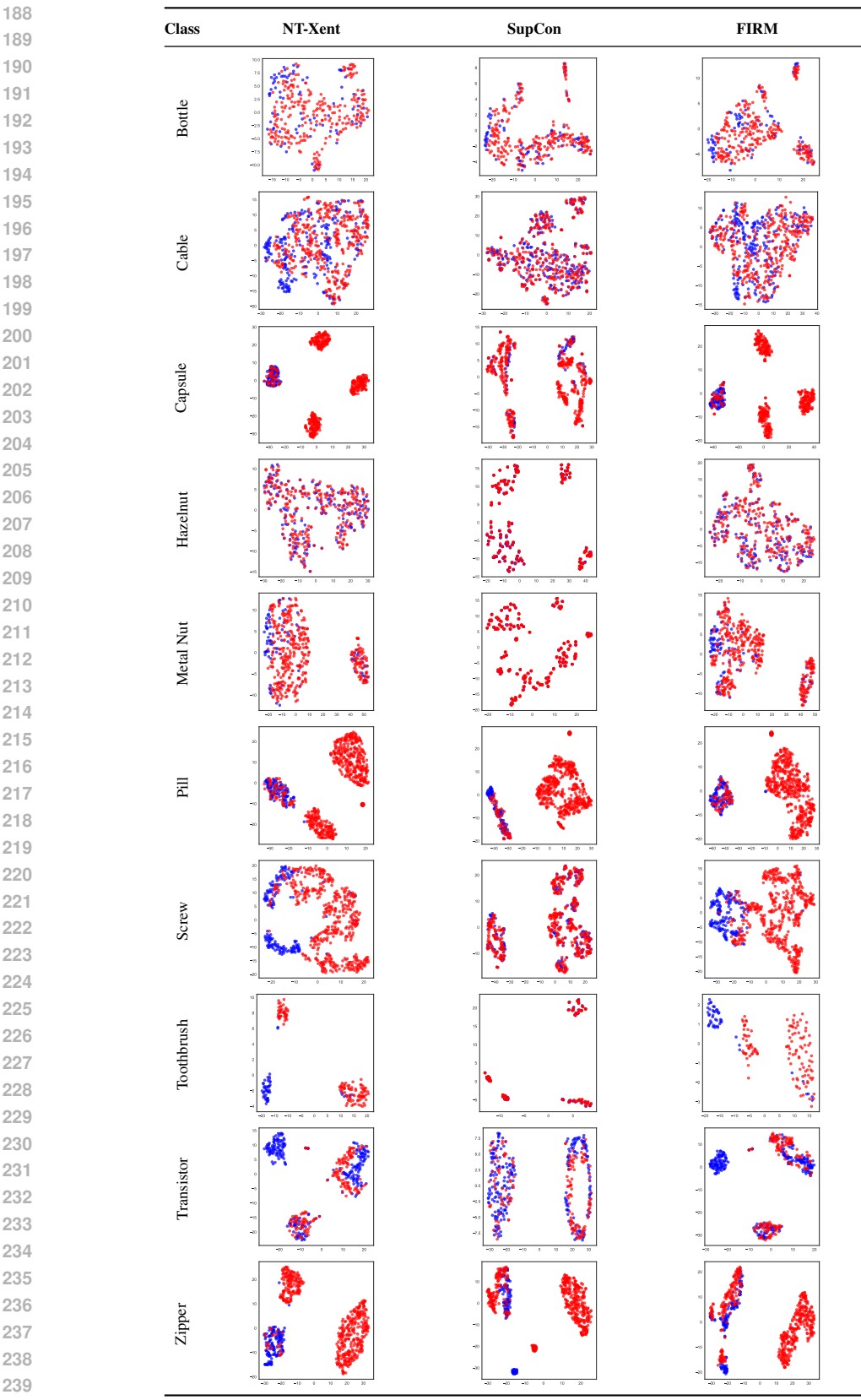

Figure 6: t-SNE plots for all object classes in the MVTec-AD dataset, visualized for NT-Xent, Sup-Con, and FIRM loss functions. Blue points indicate normal samples and red points indicate anomalies.

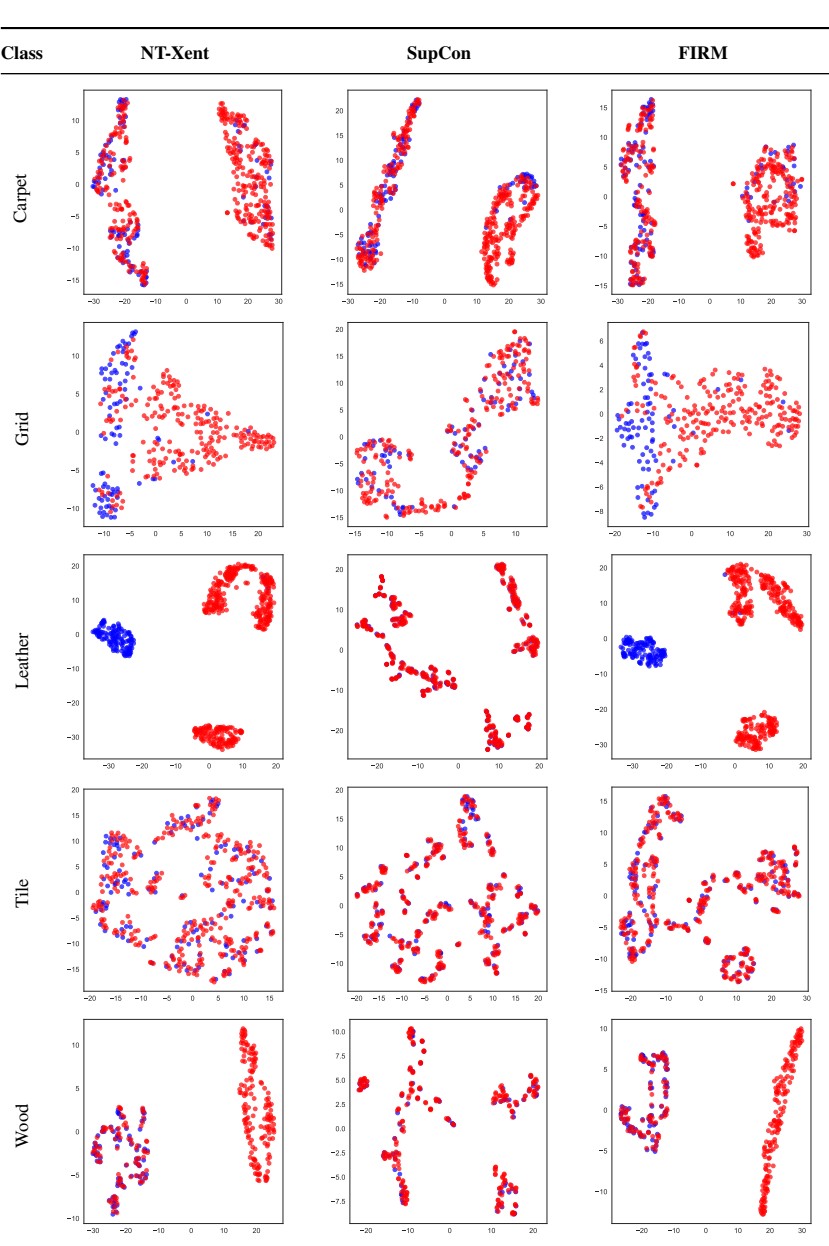

Figure 7: t-SNE plots for all texture classes in the MVTec-AD dataset, visualized for NT-Xent, SupCon, and FIRM loss functions. Blue points indicate normal samples and red points indicate anomalies.

