# OpenReview forum: "Learning In-Distribution Representations for Anomaly Detection"
_ICLR.cc/2025/Conference — ICLR 2025 Conference Withdrawn Submission_

### Official Review · Reviewer_6Xb9 · 2024-10-28

**Soundness:** 2
**Presentation:** 3
**Contribution:** 2
**Rating:** 5
**Confidence:** 4

**Summary:**

This paper indeed combines virtual OOD (out-of-distribution) samples and contrastive learning. From the main contributions and technical details, we can observe the following:

Virtual OOD and Anomaly Detection: The paper utilizes synthetic outliers, which are virtual samples generated through transformations like rotations applied to the original distribution (ID data). Alternatively, the outliers can be obtained from external datasets using Outlier Exposure (OE). These virtual samples serve as "hard negatives" to enhance the model’s ability to distinguish between normal and anomalous data​.

Contrastive Learning and the FIRM Method: The paper introduces a novel multi-positive contrastive loss called FIRM (Focused In-distribution Representation Modeling). This is an extension of traditional contrastive learning, designed to address the class collision problem. FIRM aligns multiple positive samples with the same ID anchor and leverages virtual outliers to improve the model’s ability to differentiate between normal and abnormal data​.

Advantages of the Method: The core of the FIRM loss function is to promote tight clustering of ID samples while maintaining diversity among synthetic outliers. This design effectively enhances anomaly detection performance and mitigates the class collision issue often encountered with standard contrastive learning objectives​.

Thus, this paper indeed employs virtual OOD and contrastive learning methods, with innovative improvements over traditional contrastive approaches. FIRM aims to enhance representation learning by tightly clustering ID samples and leveraging synthetic outliers to improve OOD detection.

Cons
1. I concern the novelty of this work.
2. The baselines are old. Could you intoduce resent baselines (2023-2024) related to OE? such as
Learning to Augment Distributions for Out-of-distribution Detection. NeurIPS 2022
3. DreamOOD (Dream the impossible: Outlier imagination with diffusion models) also considers to generate fake OOD data. Do you have considered this case?

**Strengths:**

Virtual OOD and Anomaly Detection: The paper utilizes synthetic outliers, which are virtual samples generated through transformations like rotations applied to the original distribution (ID data). Alternatively, the outliers can be obtained from external datasets using Outlier Exposure (OE). These virtual samples serve as "hard negatives" to enhance the model’s ability to distinguish between normal and anomalous data​.

Contrastive Learning and the FIRM Method: The paper introduces a novel multi-positive contrastive loss called FIRM (Focused In-distribution Representation Modeling). This is an extension of traditional contrastive learning, designed to address the class collision problem. FIRM aligns multiple positive samples with the same ID anchor and leverages virtual outliers to improve the model’s ability to differentiate between normal and abnormal data​.

Advantages of the Method: The core of the FIRM loss function is to promote tight clustering of ID samples while maintaining diversity among synthetic outliers. This design effectively enhances anomaly detection performance and mitigates the class collision issue often encountered with standard contrastive learning objectives​.

Thus, this paper indeed employs virtual OOD and contrastive learning methods, with innovative improvements over traditional contrastive approaches. FIRM aims to enhance representation learning by tightly clustering ID samples and leveraging synthetic outliers to improve OOD detection.

**Weaknesses:**

Cons
1. I concern the novelty of this work.
2. The baselines are old. Could you intoduce resent baselines (2023-2024) related to OE? such as
Learning to Augment Distributions for Out-of-distribution Detection. NeurIPS 2022
3. Please consider more recent OOD methods as baselines.
4. DreamOOD (Dream the impossible: Outlier imagination with diffusion models. NeurIPS 2023) also considers to generate fake OOD data. Do you have considered this case?

**Questions:**

See Weakness

---

> ### Author Response · Authors · 2024-11-22
>
> We thank the reviewer for their helpful feedback. Please find our responses below:
>
> > **Remark 1:** I concern the novelty of this work.
>
> Our work contributes to the field of anomaly detection by formally defining the characteristics of an optimal encoder for this task and designing a contrastive learning objective that aligns with these characteristics, without relying on complex data augmentations.
>
> To substantiate our claims, we conduct extensive ablation studies, rigorously evaluating the FIRM multi-positive contrastive loss against NT-Xent and SupCon in both binary and multiclass settings. These experiments offer a comprehensive understanding of our method's performance across diverse configurations.
>
> Furthermore, we have made our code publicly available to ensure transparency, and reproducibility, and to provide a foundation for further advancements by the research community.
>
> > **Remark 2:** The baselines are old. Could you intoduce resent baselines (2023-2024) related to OE? such as Learning to Augment Distributions for Out-of-distribution Detection. NeurIPS 2022. Please consider more recent OOD methods as baselines.
>
> Our work addresses mainly anomaly detection but we expand experiments to include the underexplored setting of unlabeled multiclass OOD detection, where the dataset is treated as in-distribution without class labels. This limits the availability of directly comparable baselines that performed experiments under these settings. Nonetheless, we include relevant and recent baselines such as Rotational Prediction (Rot. Pred.), DROC [1], and SSD [2].
>
> The suggested work (Learning to Augment Distributions for Out-of-Distribution Detection, NeurIPS 2022) assumes labeled OOD samples, which introduces additional information and alters the benchmark significantly, making direct comparison with our method less meaningful.
>
> We have updated Section 4 to include your suggestion [3] while emphasizing that our focus remains on anomaly detection and related tasks such as the unlabeled multiclass OOD setting.
>
> [1] Kihyuk Sohn, Chun-Liang Li, Jinsung Yoon, Minho Jin, and Tomas Pfister. Learning and evaluating representations for deep one-class classification. In International Conference on Learning Representations, 2021.
>
> [2] Vikash Sehwag, Mung Chiang, and Prateek Mittal. SSD: A unified framework for self-supervised outlier detection. In International Conference on Learning Representations, 2021.
>
> [3] Qizhou Wang, Zhen Fang, Yonggang Zhang, Feng Liu, Yixuan Li, and Bo Han. Learning to augment distributions for out-of-distribution detection. In NeurIPS, 2023.
>
> > **Remark 3:** DreamOOD (Dream the impossible: Outlier imagination with diffusion models. NeurIPS 2023) also considers to generate fake OOD data. Do you have considered this case?
>
> We acknowledge the potential of advanced methods like DreamOOD to enhance anomaly detection through sophisticated synthetic OOD data generation. However, our work focuses specifically on the training objective and evaluating contrastive learning objectives such as FIRM, NT-Xent, and SupCon under the same experimental conditions.
>
> Our aim is to rigorously compare these objectives in a controlled setting rather than exploring how different methods of generating synthetic outliers might impact performance. This approach allows us to isolate and analyze the effects of the training objective itself.

---

> > ### Comment · Reviewer_6Xb9 · 2024-11-23
> > **To response**
> >
> > After reading your responses, I maintain my score.

---

### Official Review · Reviewer_p6Sw · 2024-10-28

**Soundness:** 3
**Presentation:** 3
**Contribution:** 1
**Rating:** 1
**Confidence:** 4

**Summary:**

This paper introduces a novel contrastive loss function for representation learning in anomaly detection. The approach incorporates multiple positive samples from the in-distribution into the contrastive objective, enabling the model to learn a more compact representation space. At the same time, it differentiates these in-distribution samples from a synthetic outlier distribution, which is created by augmenting in-distribution data with rotations.

**Strengths:**

- The paper presents a clear intuition behind the proposed approach, making it easy to follow.
- Extensive experiments and figures support the main claims.

**Weaknesses:**

The objective in this paper is identical to $L_{UniCLR}$ from UniCON [1], and the anomaly score function matches that proposed in CSI [2]. Moreover, UniCON has already extended this approach to incorporate outlier exposure. Consequently, the contribution of this paper remains unclear.

[1] Wang, Guodong, et al. "Unilaterally aggregated contrastive learning with hierarchical augmentation for anomaly detection." Proceedings of the IEEE/CVF International Conference on Computer Vision. 2023.

[2] Tack, Jihoon, et al. "Csi: Novelty detection via contrastive learning on distributionally shifted instances." Advances in neural information processing systems 33 (2020): 11839-11852.

**Questions:**

- Given that the proposed method is nearly identical to the previously published UniCON approach, could the authors clarify what distinguishes their work? Are there specific improvements, modifications, or novel insights that were not covered in UniCON?
- Could the authors further explain how their method advances beyond the techniques presented in UniCON and CSI, especially regarding the objective function and anomaly score?
- Since UniCON already extended this objective to incorporate outlier exposure, in what ways does this paper expand on or improve that application?

---

> ### Author Response · Authors · 2024-11-22
>
> We thank the reviewer for the important concerns raised regarding the similarities between our proposed method and those outlined in UniCON and CSI,
> and appreciate the opportunity to clarify the distinct contributions of our work. In light of the review, we uploaded a revised manuscript to reference UniCon in Section 4 (highlighted in yellow).
>
> Our work focuses on defining the characteristics of an optimal encoder for anomaly detection
> and aligning a contrastive learning objective with these characteristics, without relying on intricate data augmentations. We have found that even without these augmentations, our method achieves comparable performance, demonstrating its robustness and efficiency.
>
> Additionally, we perform extensive ablation studies to rigorously test the FIRM objective against NT-Xent and SupCon in both binary and multiclass settings.
> These studies provide a deeper understanding of how our method performs across a range of configurations to substantiate our claims, an aspect not covered
> in the UniCON paper.
>
> Our code is made publicly available, ensuring that our experiments are transparent and reproducible to support the integrity of our findings and allow the
> community to build upon our work.
>
> Morever, it is important to clarify that the anomaly score function used in our study, derived from the CSI work, is not where our novelty lies.
> We intentionally adopted the established CSI score function and reference it in our work due to its proven effectiveness, allowing us to focus our
> innovative efforts on developing and testing our contrastive loss function.
>
> Due to the rapid progression of our field, it is a significant challenge to stay current with all emerging literature. We are committed to ensuring our
> research contributes meaningfully to the community. We hope this response adequately addresses your concerns and illustrates the independent value of our research. We are prepared to provide any further information required and look forward to the possibility of advancing our contribution
> to the anomaly detection field.

---

> > ### Comment · Reviewer_p6Sw · 2024-11-24
> >
> > Thank you for your response and for making the code publicly available. While I appreciate the clarifications and references to the ablation studies around NT-Xent and SupCon, I find the methodological contributions not distinct enough from prior work and will maintain my initial score.

---

### Official Review · Reviewer_kutR · 2024-11-01

**Soundness:** 3
**Presentation:** 4
**Contribution:** 2
**Rating:** 5
**Confidence:** 3

**Summary:**

The paper introduces the "Focused In-distribution Representation Modeling (FIRM)" loss, built on contrastive learning using multi-positive aiming to improve representation quality to prevent class collision and tackling intraclass variance problem, using synthetic outliers (by augmenting normal data) and outlier exposure method to achieve higher performance.

**Strengths:**

- Introducing a multi-positive contrastive for anomaly detection, addressing class collision and intraclass variance.
- The paper provides extensive experiments on benchmark datasets, and it employs multiple scoring functions and ablation studies, which add rigor to the evaluation, and I appreciate the hard works.
- The paper’s use of ablation studies to dissect the performance impact of each component, like scoring functions, effectively supports the design choices.

**Weaknesses:**

- In Figure 1 authors depicting loss landscapes, but these visualizations are not sufficiently clear regarding class collision and representation compactness. Additionally, a t-SNE visualization comparing FIRM and baseline representations (like CSI method) would provide concrete evidence of FIRM’s impact on clustering and separation in the representation space.
- The experiments are conducted on typical anomaly detection benchmarks like CIFAR-10, SVHN and etc., which are well-studied but may not adequately represent the complexities of real-world anomaly detection tasks because of: First the resolution of these datasets is very low and models like ResNet-18 could easily be overfitted on them (as you have a train model with 2000 epochs!). It is recommended apply your method on datasets like MvTech-AD and VisA for anomaly detection and ImageNet-30 for OOD.
- While the paper introduces the multi-positive objective, it lacks a rigorous theoretical analysis to justify why this approach is optimal for anomaly detection.

**Questions:**

1- Does the author test the generalization of the provided method on augmentations and noises?

2- In provided "Ensemble score" you use data and 3 rotations of it as an input and then add random crop to it. Wouldn't blindly rotating an image like a car made it anomaly and it will affect your score? Then randomly cropping and image also could hurt detection of anomaly by removing anomaly part of the object.

3- In your experiment for creating synthetic outliers is again above argument will apply by creating an anomaly data using its rotation (like a rotated ship). I know that CSI with also used this type of rotation as their main contribution for achieving a good representation, but their works lacks the generality, and the accuracy will drop over some augmentation. Did you even test your method for generality?

---

> ### Author Response · Authors · 2024-11-22
>
> We appreciate the reviewer’s recognition of the rigorous evaluation of the proposed contribution. Please find our responses below:
>
> > **Remark 1:** In Figure 1 authors depicting loss landscapes, but these visualizations are not sufficiently clear regarding class collision and representation compactness. Additionally, a t-SNE visualization comparing FIRM and baseline representations (like CSI method) would provide concrete evidence of FIRM’s impact on clustering and separation in the representation space.
>
> To better demonstrate the impact of FIRM on class collision and representation compactness, we included t-SNE visualizations in our revised manuscript comparing
> FIRM to baseline representations SupCon and NT-Xent to support our claim regarding FIRM's effectiveness in promoting clustering
> and separation in the representation space. Visualizations are included in the revised manuscript in Section 3 and Appendix E.
>
> > **Remark 2:** The experiments are conducted on typical anomaly detection benchmarks like CIFAR-10, SVHN and etc., which are well-studied but may not adequately represent the complexities of real-world anomaly detection tasks because of: First the resolution of these datasets is very low and models like ResNet-18 could easily be overfitted on them (as you have a train model with 2000 epochs!). It is recommended apply your method on datasets like MvTech-AD and VisA for anomaly detection and ImageNet-30 for OOD.
>
> We conducted additional experiments with the MVTec AD dataset for anomaly detection, featuring higher resolution real-world anomalies. Results have been added
> to the revised manuscript Section 3.1 (highlighted in yellow). The code for these experiments has been updated and is now available in our repository.
>
> > **Remark 3:** While the paper introduces the multi-positive objective, it lacks a rigorous theoretical analysis to justify why this approach is optimal for anomaly detection.
>
> The multi-positive objective is designed to address the homogeneity typically observed in normal data in anomaly detection contexts. By encouraging a
> compact representation of in-distribution (ID) data and a diverse representation of synthetic outliers, our approach aims to enhance the model's sensitivity
> to anomalies. The rationale is that by diversifying the representation of outliers while maintaining the compactness of the ID samples, the model can better
> discriminate between normal and anomalous instances.
>
> To empirically test this theory, we conducted ablation studies comparing the Supervised Contrastive Learning (SupCon) loss in both binary and multiclass settings,
> as detailed in Table 5 in the appendix of our manuscript. In the binary setting, SupCon encourages compactness for both classes (ID and synthetic outliers) which can sometimes lead to representational collapse. This was particularly evident in the performance dips observed in specific classes such as "Bird" and "Cat,"
> where the scores were significantly lower compared to the multiclass setting.
>
> Conversely, in the multiclass setting (SupCon*), where synthetic outliers are labeled according to their rotation angles (90, 180, and 270 degrees as
> classes 1, 2, and 3, respectively), we observed a substantial improvement in anomaly detection performance. This setting effectively prevents the representational
> collapse observed in the binary setting by enhancing separability among synthetic outlier representations.
>
> These results indicate that promoting diversity among synthetic outliers, in addition to compactness of ID samples, significantly enhances anomaly detection.
> Furthermore, FIRM loss does not require labels for synthetic outliers, making it a versatile tool for real-world applications where labeling can be impractical.
>
> > **Remark 4:**  Does the author test the generalization of the provided method on augmentations and noises?
>
> Could you please clarify if your question pertains to our general approach to testing robustness against various perturbations?
> Any additional details would greatly assist us in providing you with a more precise response.

---

> ### Author Response · Authors · 2024-11-22
>
> Continuation.
>
> > **Remark 5:**  In provided "Ensemble score" you use data and 3 rotations of it as an input and then add random crop to it. Wouldn't blindly rotating an image like a car made it anomaly and it will affect your score? Then randomly cropping and image also could hurt detection of anomaly by removing anomaly part of the object.
>
> The use of rotations and crops can potentially alter the distribution characteristics of certain objects, where such transformations might not
> shift the distribution sufficiently. In our experiments, we acknowledge that certain classes, particularly those involving vehicles like "Plane" and "Car"
> may not benefit significantly from rotations as synthetic outliers. This is because these transformations do not introduce sufficient perceptual differences,
> leading to potential class collisions. Our findings in Table 9 (a) in the revised manuscript, discussed in Appendix D, illustrate that while FIRM consistently outperforms other methods
> like NT-Xent and SupCon in complex classes, it shows relatively lower performance in "Plane" and "Car" classes. This phenomenon can be attributed to the overlap
> between ID samples and hard negatives generated through these rotations, which may not be distinct enough to prevent representational collapse.
>
> However, it is also crucial to note that the use of rotations helps the model learn to generalize from these angles, essentially training it to recognize and
> extract relevant features. The use of "hard" augmentations like rotation can actually enhance performance by acting as hard negatives
> that push augmented samples away from the original sample, thus providing a more robust learning experience [1]. This aligns with our findings in Table 6,
> where the "s_ens" score for FIRM with rotations significantly outperforms other methods, illustrating the effectiveness of our approach in leveraging these
> hard negatives.
>
> As for random cropping, it is a valid concern that too aggressive cropping might remove critical features of anomalies, thereby impairing the detection capability.
> In our implementation, we adhere to the best practices established in previous studies [1], where the cropping range must be carefully controlled.
> We ensure that our cropping parameters are limited in range (scaling between 0.5 to 1) to retain essential features while still providing variability.
>
> > **Remark 6:**  In your experiment for creating synthetic outliers is again above argument will apply by creating an anomaly data using its rotation (like a rotated ship). I know that CSI with also used this type of rotation as their main contribution for achieving a good representation, but their works lacks the generality, and the accuracy will drop over some augmentation. Did you even test your method for generality?
>
> In our experiments, the utilization of rotations as synthetic outliers is employed to serve as hard negatives, which effectively help shift the
> distribution between positive and negative samples [1]. This strategy allows the model to discern more rigorously between in-distribution and
> out-of-distribution samples, thereby improving generalizability. The objective of using such "hard" augmentations, like rotations, is to increase
> the robustness of the model to a variety of anomalies through the subsequent shift in the distribution between positive and negative samples, which is
> evidenced by our generalized performance across different datasets and augmentation scenarios with and without rotations (See Table 1 FIRM and FIRM w/ OE).
>
> [1] Jihoon Tack, Sangwoo Mo, Jongheon Jeong, and Jinwoo Shin. Csi: Novelty detection via contrastive learning on distributionally shifted instances. In Advances in Neural Information Processing Systems, volume 33, pp. 11839–11852. Curran Associates, Inc., 2020.

---

> ### Comment · Reviewer_kutR · 2024-11-25
>
> Thank you for your effort, addition tests and plots. The additional experiments on MVTec AD provide valuable evidence of real-world applicability, addressing concerns about the limitations of low-resolution datasets. Despite the better results on MvTecAD dataset compared to NT-Xent and SupCon the t-SNE plot did not show any improvement in representation separability.
>
> I appreciate the clarifications on generality and synthetic outliers, though I encourage exploring alternative augmentations to mitigate potential limitations with structurally similar anomalies.

---

> > ### Author Response · Authors · 2024-11-27
> >
> > Thank you for your follow-up.
> >
> > While the t-SNE plots may not show a clear difference in every case, there are notable instances where we observe greater compactness of normal representations and improved separability from anomalies, particularly in the comparison between SupCon (binary) and FIRM (see the full t-SNE plots for objects in Figure 6 and textures in Figure 7). Basing conclusions entirely on the visual assessment of these t-SNE plots might not capture the full extent of the improvements; however, this is clearly reflected in the quantitative outcomes, as FIRM achieves better performance across most classes (see Tables 12 and 13 for the full results).
> >
> > Moreover, our decision score, based on cosine similarity to the training set, directly evaluates representation quality without relying on additional non-linear classifiers. While this metric may not explicitly indicate increased compactness of in-distribution, it reflects better separability between in-distribution and anomalies, as shown by the higher AUROC scores.
> >
> > We hope these clarifications and additional results address your concerns and provide further insight into the strengths of our approach. Please feel free to let us know if there are any remaining questions or areas where further explanation could be helpful.
> >
> > Thank you once again for your thoughtful review and valuable feedback!

---

### Official Review · Reviewer_N64u · 2024-11-02

**Soundness:** 2
**Presentation:** 3
**Contribution:** 2
**Rating:** 3
**Confidence:** 5

**Summary:**

The paper proposed a novel contrastive loss for the anomaly detection, which favors representation invariance for the inliers, but not for the (synthetic) outliers (Eq. 2). This is expected to make the inliers or normal samples compact in the learned feature space, while allowing the outliers be diverse in that space. The proposed method is tested in multiple datasets, such as CIFAR-10, 100, Fashion-MNIST, Cats-vs-Dogs in the anomaly detection setting where one class is considered as normal and the others as anomaly. They also considered testing the proposed method in the OOD detection, where an entirely different dataset is considered as OOD, e.g. CIFAR-10 vs SVHN. The results showed minor improvements over the baselines.

**Strengths:**

The idea to highlight the ID in the positive pairs in contrastive learning for anomaly detection sounds interesting and novel. The results are complemented by several ablations against vanilla NT-Xent, and SupCon contrastive losses.

**Weaknesses:**

- The improvement margin is very narrow compared to the competitors such as CSI. For instance in table 1, the AUROC is improved by only 1.1% points compared to the CSI (s_ens). The other variant of the method that relies on OE, is not fair when experimenting CIFAR-10 or 100, as the OE has shown to perform well when its distribution is close to the ID (e.g. see "RODEO: Robust Outlier Detection via Exposing Adaptive Out-of-Distribution Samples" by H. Mirzaei et al ICML 2024).
- Comparison against standard anomaly detection datasets, such as MVTec AD, and VisA, or medical datasets are missing. This makes validation of the claims universality difficult.
- If the author claims are true, we expect the false positive, i.e. wrongly detecting a normal sample as anomaly, should be high. Because if the normal samples lack representational invariance, it would be more likely for the model to make mistakes on those cases. However, no evidence is provided along this direction.
- In the OOD detection application, the proposed method (without OE) fails to outperform CSI in harder cases of distinguishing CIFAR-10 from the near-distribution datasets such as CIFAR-100 and ImageNet. The improvement only happens when a huge OE is used, which is unfair as mentioned in the first point earlier.

**Questions:**

- How does your method perform when tested against MVTec AD, or medical datasets?
- Why the improvement margin is so small compared to the CSI?
- How does the FPR @ TPR = 95% looks like for your model compared to the other contrastive learning based models?

---

> ### Author Response · Authors · 2024-11-22
>
> We thank the reviewer for the thorough feedback and critical points raised. Please find our responses below:
>
> > **Remark 1:** The improvement margin is very narrow compared to the competitors such as CSI. For instance in table 1, the AUROC is improved by only 1.1% points compared to the CSI (s_ens).
>
> The improvement margin may seem narrow when compared to CSI, however, it is important to note that this claim holds true in the case that we allow CSI
> a substantial training effort of 2000 epochs. FIRM achieves similar performance within the first few epochs, as seen in Figure 3.
>
> > **Remark 2:** The other variant of the method that relies on OE, is not fair when experimenting CIFAR-10 or 100, as the OE has shown to perform well when its distribution is close to the ID (e.g. see "RODEO: Robust Outlier Detection via Exposing Adaptive Out-of-Distribution Samples" by H. Mirzaei et al ICML 2024).
>
> While we understand the concern regarding overall performance impact of using OE to more closely resembling distributions in the CIFAR datasets,
> our main objective lies in positioning the performance of FIRM loss against NT-Xent and Supcon across the same experimental conditions to evaluate
> their respective performances against each other. We revised our manuscript to address this in Section 2 (text highlighted in yellow), referencing the findings in
> 'RODEO: Robust Outlier Detection via Exposing Adaptive Out-of-Distribution Samples'.
>
> > **Remark 3:** Comparison against standard anomaly detection datasets, such as MVTec AD, and VisA, or medical datasets are missing. This makes validation of the claims universality difficult.
>
> We performed additional experiments with the MvTec-AD dataset and included the results in the revised manuscript in Section 3.1. The code for these experiments has been updated and is now available in our repository.
>
> > **Remark 4:** If the author claims are true, we expect the false positive, i.e. wrongly detecting a normal sample as anomaly, should be high. Because if the normal samples lack representational invariance, it would be more likely for the model to make mistakes on those cases. However, no evidence is provided along this direction.
>
> FIRM objective encourages compact clustering of ID representations, which reduces ambiguity in distinguishing ID from OOD samples. This approach helps to minimize false positives rather than increasing them. The high AUROC scores reported in our experiments reflect a low false positive rate, as AUROC evaluates the balance between true positives and false positives.
>
> To provide further clarity, our theoretical discussions in Section 2 explicitly define the properties of an idealized encoder, which we drew inspiration from to design our objective.
>
> The goal of our design is to improve ID compactness without causing representational collapse. Collapse would occur if the encoder mapped both ID and OOD samples to the same point, which would severely limit the model's ability to distinguish between normal and anomalous data. This distinction is key to understanding why the FIRM objective does not increase false positives.
>
> > **Remark 5:** In the OOD detection application, the proposed method (without OE) fails to outperform CSI in harder cases of distinguishing CIFAR-10 from the near-distribution datasets such as CIFAR-100 and ImageNet. The improvement only happens when a huge OE is used, which is unfair as mentioned in the first point earlier.
>
> It is important to highlight that while FIRM primarily aims to address scenarios involving homogeneous normal data, it also demonstrates robust performance
> in more challenging settings. Specifically, we performed experiments using a "fixed" version of ImageNet (See Table 3), which can be considered a near-OOD dataset due to its
> closer distribution to the ID. In these cases, FIRM outperforms baseline methods, emphasizing its capability beyond the initially targeted application scope.
> We consider these findings as additional evidence of the method’s robustness rather than the core results, extending the applicability of FIRM to broader contexts.
>
> > **Remark 6:** How does your method perform when tested against MVTec AD, or medical datasets?
>
> See answer to remark 3.
>
> > **Remark 7:** Why the improvement margin is so small compared to the CSI?
>
> See answer to remark 1.
>
> > **Remark 8:** How does the FPR @ TPR = 95% looks like for your model compared to the other contrastive learning based models?
>
> We acknowledge that FPR @ TPR = 95% is useful for evaluating the trade-off between sensitivity and false alarms at a specific operating point. However, we used standard metrics like AUROC, which comprehensively assess the balance between true and false positives across thresholds and enable consistent comparisons with other methods. The high AUROC scores achieved by our method already highlight its robustness in minimizing false positives while maintaining strong anomaly detection performance.

---

> ### Author Response · Authors · 2024-11-27
>
> Dear Reviewer N64u,
>
> Thank you for your thoughtful feedback on our manuscript. We’ve carefully addressed your comments and would greatly appreciate any additional thoughts or suggestions you might have.
>
> Your input is invaluable, and we are happy to make further revisions based on your recommendations.
>
> Sincerely,
> The Authors

---

> > ### Comment · Reviewer_N64u · 2024-12-01
> >
> > Thanks the authors for their detailed rebuttal. After reading through it, I still believe that the improvement with respect to CSI is marginal, and the long training of CSI is not a major drawback. In addition, MVTec results do not look promising, despite NSA performing dataset specific augmentations, as these augmentations are available to any method. For these reasons, I tend to keep my score unchanged.

---

### Official Review · Reviewer_Ysdm · 2024-11-02

**Soundness:** 3
**Presentation:** 3
**Contribution:** 2
**Rating:** 5
**Confidence:** 4

**Summary:**

This study introduces FIRM (Focused In-distribution Representation Modeling), a novel contrastive approach for anomaly detection. Traditional methods struggle with high-dimensional data and class collision, especially with homogeneous datasets. FIRM addresses these issues by compacting in-distribution representations and separating them from synthetic outliers. It outperforms existing methods in anomaly detection benchmarks, improving robustness and representation quality, especially in ensemble settings and with Outlier Exposure (OE).

**Strengths:**

* This paper addresses an important problem of learning representation for anomaly detection, which is challenging due to the lack of rich training signals in typical in-distribution datasets.
* The proposed method is straightforward and shows promising results on many benchmarks.

**Weaknesses:**

* While the experimental results are promising, the technical novelty of the proposed algorithm is somewhat limited, as it essentially represents a minor modification of class labels in the conventional contrastive learning objective.
* The paper provides limited theoretical discussion.
* Most experiments are conducted with relatively small images (32x32 pixels), and it is unclear whether the cats-vs-dogs dataset is also resized to this dimension. Practical anomaly detection scenarios often involve larger images, which may lead to qualitatively different outcomes and trends when experimented with. It would be beneficial if the paper included experiments with industrial anomaly detection tasks or images larger than 128x128 pixels.

**Questions:**

* Could you provide the details on how the images are preprocessed in the experiments?

---

> ### Author Response · Authors · 2024-11-22
>
> We thank the reviewer for the thorough feedback and critical points raised. Please find our responses below:
>
> > **Remark 1:** While the experimental results are promising, the technical novelty of the proposed algorithm is somewhat limited, as it essentially represents a minor modification of class labels in the conventional contrastive learning objective.
>
> We introduce a straightforward yet effective modification to the contrastive learning objective, addressing a critical but often neglected aspect of anomaly detection [1, 2]. Although our approach may seem simple, it is effective, as illustrated in Figure 3. FIRM achieves comparable mean AUC performance within just 100 epochs, a result that NT-Xent and SupCon (multiclass) typically require 2000 epochs to achieve. Additionally, our ablation studies in Figure 4 underscore the critical role of promoting distinct separations in the representations of synthetic outliers.
>
> [1] Tal Reiss and Yedid Hoshen. Mean-shifted contrastive loss for anomaly detection. In Proceedings of the AAAI Conference on Artificial Intelligence, volume 37, pp. 2155–2162, 2023.
>
> [2] Kihyuk Sohn, Chun-Liang Li, Jinsung Yoon, Minho Jin, and Tomas Pfister. Learning and evaluating representations for deep one-class classification. In International Conference on Learning Representations, 2021.
>
> > **Remark 2:** The paper provides limited theoretical discussion.
>
> We expanded the theoretical discussion in Section 2 of the paper (highlighted in yellow).
>
> > **Remark 3:** Most experiments are conducted with relatively small images (32x32 pixels), and it is unclear whether the cats-vs-dogs dataset is also resized to this dimension. Practical anomaly detection scenarios often involve larger images, which may lead to qualitatively different outcomes and trends when experimented with. It would be beneficial if the paper included experiments with industrial anomaly detection tasks or images larger than 128x128 pixels.
>
> In our work, we followed standard experimental protocols widely used in the field [1, 2] to enable a fair comparison with related methods. Specifically:
>
> For the Cats-vs-Dogs dataset, we followed the image processing setup outlined by Sohn et al. [2] as described in Section 3.1 of our paper. The images were resized to 64x64 pixels.
> For CIFAR-10, CIFAR-100, and Fashion-MNIST datasets, the images were resized to 32x32 pixels to maintain consistency with established baselines.
> We incorporated these clarifications in the revised manuscript in Section 3.1 (highlighted in yellow).
>
> To address the broader applicability of our method to industrial anomaly detection tasks with larger images, we conducted additional experiments
> using the MVTec-AD dataset added to Section 3.1 of our revised manuscript under 'Defect anomaly detection' (highlighted in yellow), allowing us to evaluate our approach in scenarios more representative of real-world industrial settings.
> Additionally, the code for these experiments has been updated and is now available in our repository.
>
> [1] Jihoon Tack, Sangwoo Mo, Jongheon Jeong, and Jinwoo Shin. Csi: Novelty detection via contrastive learning on distributionally shifted instances. In Advances in Neural Information Processing Systems, volume 33, pp. 11839–11852. Curran Associates, Inc., 2020.
> [2] Kihyuk Sohn, Chun-Liang Li, Jinsung Yoon, Minho Jin, and Tomas Pfister. Learning and evaluating representations for deep one-class classification. In International Conference on Learning Representations, 2021.
>
> > **Remark 4:** Could you provide the details on how the images are preprocessed in the experiments?
>
> See answer to remark 3.

---

> > ### Comment · Reviewer_Ysdm · 2024-11-26
> > **Thanks for response.**
> >
> > I appreciate the detailed comment. However, my concern about the technical novelty remains, even after reading the responses of other reviewers. I would like to keep my original score.

---

### Note · Authors · 2025-01-06

**Comment:**

We want to thank the reviewers for their invaluable feedback on our submission. After careful consideration, we have decided to withdraw our paper and resubmit.

We would also like to highlight our proposed pretext task and loss function for anomaly detection within contrastive settings. We argue that FIRM is a more suitable pretext task for anomaly detection in semi-supervised settings, where the in-distribution is known to consist solely of normal samples compared to NT-Xent and SupCon. Encouraging compact representations between samples of a given class is a familiar concept, as implemented by SupCon; similarly, FIRM encourages compact in-distribution representations. Additionally, given that synthetic outliers can be seen as a set of unlabeled samples, FIRM leverages the NT-Xent pretext task, leveraging self-labeling for these samples while also using them as negatives to the in-distribution samples. This not only promotes separation between the in-distribution and synthetic outliers but also encourages distinct separations among the synthetic outliers themselves. Our experiments confirm the effectiveness of FIRM, which converges up to 40 times faster than NT-Xent and 20 times faster than SupCon in binary settings and achieves superior results.

**Withdrawal Confirmation:**

I have read and agree with the venue's withdrawal policy on behalf of myself and my co-authors.